# The mechanism of mammalian proton-coupled peptide transporters

**Simon M Lichtinger[1,2], Joanne L Parker[1,2], Simon Newstead[1,2]\*, Philip C Biggin[1]\***

[1]Structural Bioinformatics and Computational Biochemistry, Department of Biochemistry, University of Oxford, Oxford, United Kingdom; [2]The Kavli Institute for Nanoscience Discovery, University of Oxford, Oxford, United Kingdom

**\*For correspondence:**
simon.newstead@bioch.ox.ac.uk (SN);
philip.biggin@bioch.ox.ac.uk (PCB)

**Competing interest:** The authors declare that no competing interests exist.

**Abstract** Proton-coupled oligopeptide transporters (POTs) are of great pharmaceutical interest owing to their promiscuous substrate binding site that has been linked to improved oral bioavailability of several classes of drugs. Members of the POT family are conserved across all phylogenetic kingdoms and function by coupling peptide uptake to the proton electrochemical gradient. Cryo-EM structures and alphafold models have recently provided new insights into different conformational states of two mammalian POTs, SLC15A1, and SLC15A2. Nevertheless, these studies leave open important questions regarding the mechanism of proton and substrate coupling, while simultaneously providing a unique opportunity to investigate these processes using molecular dynamics (MD) simulations. Here, we employ extensive unbiased and enhanced-sampling MD to map out the full SLC15A2 conformational cycle and its thermodynamic driving forces. By computing conformational free energy landscapes in different protonation states and in the absence or presence of peptide substrate, we identify a likely sequence of intermediate protonation steps that drive inward-directed alternating access. These simulations identify key differences in the extracellular gate between mammalian and bacterial POTs, which we validate experimentally in cell-based transport assays. Our results from constant-PH MD and absolute binding free energy (ABFE) calculations also establish a mechanistic link between proton binding and peptide recognition, revealing key details underpining secondary active transport in POTs. This study provides a vital step forward in understanding proton-coupled peptide and drug transport in mammals and pave the way to integrate knowledge of solute carrier structural biology with enhanced drug design to target tissue and organ bioavailability.

## eLife assessment

This study provides **important** insight into the mechanisms of proton-coupled oligopeptide transporters. It uses enhanced-sampling molecular dynamics (MD), backed by cell-based assays, revealing the importance of protonation of selected residues for PepT2 function. The simulation approaches are **convincing**, using long MD simulations, constant-pH MD and free energy calculations. Overall, the work has led to findings that will appeal to structural biologists, biochemists, and biophysicists studying membrane transporters.

## Introduction

Cells require an external lipid membrane to separate their internal cytoplasm from the environment. Since the membrane permeability of common solutes spans ten orders of magnitude, some molecules diffuse readily across the membrane, while the translocation of others requires facilitation by carriers (*Stillwell, 2016*; *Pizzagalli et al., 2021*). Understanding the processes by which small molecules cross membranes is of key pharmacological interest owing to their role in drug delivery, which may be mediated by passive diffusion, protein carriers, or a combination of both (*Sugano et al., 2010*). The

**eLife digest** The cells in our body are sealed by a surrounding membrane that allows them to control which molecules can enter or leave. Desired molecules are often imported via transport proteins that require a source of energy. One way that transporter proteins achieve this is by simultaneously moving positively charged particles called protons across the membrane.

Proteins called POTs (short for proton-coupled oligopeptide transporters) use this mechanism to import small peptides and drugs in to the cells of the kidney and small intestine. Sitting in the centre of these transporters is a pocket that binds to the imported peptide which has a gate on either side: an outer gate that opens towards the outside of the cell, and an inner gate that opens towards the cell's interior. The movement of protons from the outer to the inner gate is thought to shift the shape of the transporter from an outwards to an inwards-facing state. However, the molecular details of this energetic coupling are not well understood.

To explore this, Lichtinger et al. used computer simulations to pinpoint where protons bind on POTs to trigger the gates to open. The simulations proposed that two sites together make up the outward-facing gate, which opens upon proton binding. Lichtinger et al. then validated these sites experimentally in cultured human cells that produce mutant POTs.

After the desired peptide/drug has attached to the binding pocket, the protons then move to two more sites further down the transporter. This triggers the inner gate to open, which ultimately allows the small molecule to move into the cell.

These findings represent a significant step towards understanding how POTs transport their cargo. Since POTs can transport a range of drugs from the digestive tract into the body, these results could help researchers design molecules that are better absorbed. This could lead to more orally available medications, making it easier for patients to adhere to their treatment regimen.

solute-carrier (SLC) superfamily encompasses 65 families of more than 450 genes, with substrates ranging in size from simple ions to complex macromolecules used in metabolism and signalling (*Pizzagalli et al., 2021*). Within this superfamily, the SLC15 family includes POTs that have significant homology through all domains of life and are evolutionarily ancient (*Daniel et al., 2006*). Of the four mammalian family members, PepT1 (SLC15A1), and PepT2 (SLC15A2) are the most well studied. The former is predominantly expressed in the small intestine and characterised as a low-affinity, high-capacity transporter (*Fei et al., 1994*). The latter has a broader expression pattern including the kidneys, lungs, and brain and is described as high-affinity, low-capacity (*Kottra and Daniel, 2004*). As secondary-active transporters, they couple uphill substrate translocation to the symport of protons down their electrochemical gradient (*Fei et al., 1994*; *Rubio-Aliaga et al., 2000*). The peptide–proton stoichiometry is not conserved between different substrates and POT family members (*Parker et al., 2014*). For PepT1, stoichiometries of 1:1 and 2:1 have been reported for neutral/basic and acidic di-peptides, respectively (*Fei et al., 1994*; *Steel et al., 1997*). For PepT2, a 2:1 stoichiometry was reported for the neutral di-peptide D-Phe-L-Ala and 3:1 for anionic D-Phe-L-Glu (*Chen et al., 1999*). Alternatively, *Fei et al., 1999* have found 1:1 stoichiometries for either of D-Phe-L-Gln (neutral), D-Phe-L-Glu (anionic), and D-Phe-L-Lys (cationic). Here, we work under the assumption of a 2:1 stoichiometry for neutral di-peptides, motivated also by our computational results that indicate distinct and additive roles played by two protons in the conformational cycle mechanism.

POT family transporters belong to the major facilitator superfamily (MFS) and share a conserved topology of two six-helix bundles that form the functional transport domain, their N-and C-termini facing the cytoplasm. (*Newstead et al., 2011*). They operate via an alternating access mechanism encoded in four inverted topology repeats, progressively reorienting the N-and C-terminal bundles to cycle through outwards-facing (OF), occluded (OCC), and inwards-facing (IF) states (*Radestock and Forrest, 2011*). Since the first structure of a POT family member was published (*Newstead et al., 2011*), many procaryotic (*Solcan et al., 2012*; *Guettou et al., 2013*; *Doki et al., 2013*; *Lyons et al., 2014*; *Guettou et al., 2014*; *Zhao et al., 2014*; *Fowler et al., 2015*; *Boggavarapu et al., 2015*; *Beale et al., 2015*; *Parker et al., 2017*; *Martinez Molledo et al., 2018*; *Ural-Blimke et al., 2019*; *Minhas and Newstead, 2019*; *Stauffer et al., 2022*; *Kotov et al., 2023*) and plant (*Parker and Newstead, 2014*; *Sun et al., 2014*) homologues have been structurally and biochemically characterised, all in IF

states with varying degrees of occlusion (see *Figure 1a* for an overview of available POT structures and their conformational states). Several residues have been suggested to be involved in proton transfer, including a partially conserved histidine on TM2 (H87; residue numbers refer to PepT2, if not specified otherwise) (*Terada et al., 1996*; *Fei et al., 1997*; *Chen et al., 2000*; *Omori et al., 2021*; *Parker et al., 2021*) and two conserved glutamates on TM1 (E53 and E56) (*Jensen et al., 2012*; *Doki et al., 2013*; *Aduri et al., 2015*), while simulations have helped our understanding of proton-transfer processes and conformational changes (*Parker et al., 2017*; *Selvam et al., 2018*; *Batista et al., 2019*; *Li et al., 2022*). However, the details of the molecular mechanism of alternating access in POTs, particularly regarding the coupling of conformational changes, substrate binding, and proton movement to each other, remain unclear.

Cryo-EM and Alphafold 2 have recently provided views of mammalian POTs in conformations spanning from OF via inward-facing-partially occluded to fully-open IF (*Parker et al., 2021*; *Killer et al., 2021*; *Shen et al., 2022*; *Jumper et al., 2021*). From these structures emerges a picture where the intracellular gate is constituted by broad close-packing of hydrophobic residues on TM 4, 5, 10, and 11, with possible stabilisation from the conserved D170–K642 salt-bridge. The extracellular gate appears to be spread along the cleft between the N-and C-terminal bundles, with contributions from the H87 (TM 2) – S321 (TM 7) polar interaction network as well as the R206 (TM 5) – D342 (TM 8) and K64 (TM 1) – D317 (TM 7) salt bridges (*Figure 1b*). This is intriguing, because the mammalian H87 residue is only conserved in some prokaryotic homologues, and R206–D342 just among mammalian POTs. We speculate based on this feature that the extracellular gating mechanism could be less conserved than POT alternating access in general. As for the intracellular gating mechanism, an involvement of the D170–K642 salt bridge has been suggested, and the OF structure shows close-packing of several hydrophobic residues (F184, Y183, F187, L630, and Y634) that constrict access to the binding site from the intracellular side (*Figure 1c*; *Parker et al., 2021*). It is not known thus far how the opening of the intracellular gate (i.e. the OCC→IF transition) is triggered, and how it is coupled to proton movement and the presence of substrate.

POTs accommodate their substrates in a highly conserved binding pocket, interfacing between an acidic patch on the C-terminal bundle and a basic patch on the N-terminal bundle (*Figure 1c*). For di-peptides, the N-terminus is coordinated by E622 (TM 10) together with N192 (TM 5) and N348 (TM 8), while the C-terminus engages R57 (TM 1, or the equivalent arginine (R27) residue in PepT1) as well as Y94 (TM 2). Another conserved tyrosine, Y61 (TM 1), hydrogen-bonds to features of the peptide backbone.(*Lyons et al., 2014*; *Martinez Molledo et al., 2018*; *Killer et al., 2021*). Tri-peptides may adopt a similar orientation as di-peptides (*Guettou et al., 2014*), or sit vertically in the transporter binding pocket (*Lyons et al., 2014*), although it has been suggested that this vertical electron density could alternatively be explained by a bound HEPES molecule (*Martinez Molledo et al., 2018*). Considering the consensus structural interaction pattern, we decided to investigate primarily the role of E622 and R57 in holding the substrate, and also note that R57 is part of the highly conserved E53xxER motif (*Figure 1d*; *Newstead, 2017*). The second glutamate in this motif (E56) in particular has been linked to proton coupling experimentally (*Jensen et al., 2012*). Since R57 interacts with both the ExxER glutamates and the substrate C-terminus, we hypothesise that it may play an important role in substrate–proton coupling.

In this study, we use extensive unbiased and enhanced-sampling MD simulations (totalling close to 1 ms of sampling) to show how changes in protonation states of H87 and D342 control the OCC↔OF transition as an extracellular gate. We validate the importance of these residues for transporter function in cell-based transport assays. We also elucidate the role of E53xxER glutamates and the substrate-engaging E622 in controlling the OCC↔IF transition, thereby identifying a clear molecular basis for the directionality of proton movement coupling to conformational changes. Furthermore, we establish several distinct effects of the presence of substrate, coupling ligand binding with protein conformational changes and also linking it to protonation of the E56 and E622 titratable residues. Taken together, our work provides for the first time a detailed model of a plausible sequence of steps for substrate and proton-coupled alternating access in mammalian POTs.

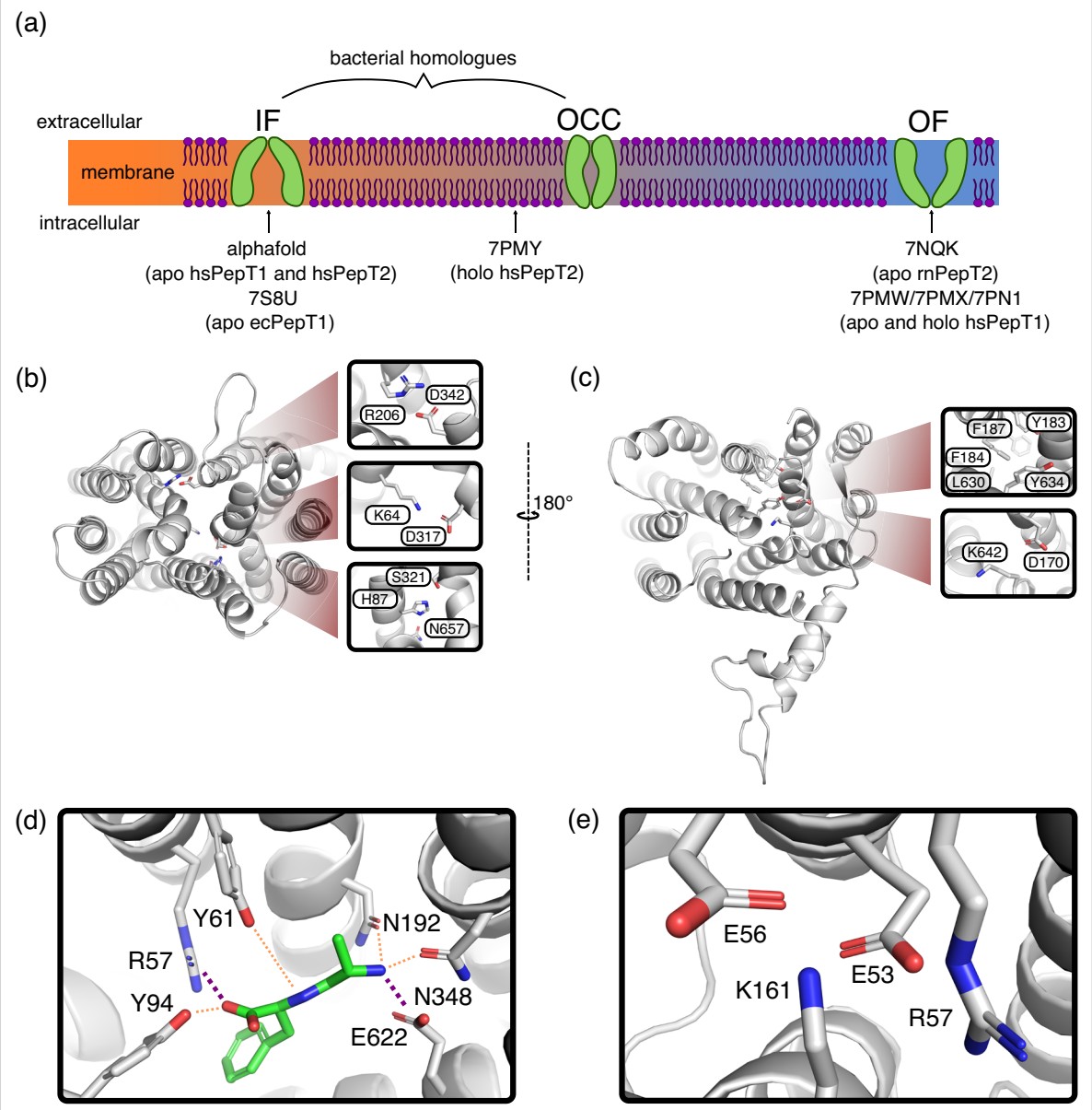

**Figure 1.** Key features of Proton-coupled oligo-peptide transporter (POT) structures. (**a**) Schematic overview of the conformational diversity of available mammalian POT structures. Intermediate positions between states indicate partial gate opening. (**b**) Alphafold-predicted inwards-facing (IF) HsPepT2 structure (top view), highlighting potential inter-bundle extracellular gate interactions. (**c**) Outwards-facing (OF) Cryo-EM structure of apo RnPepT2 (7NQK, bottom view) (*Parker et al., 2021*), highlighting potential inter-bundle intracellular gate interactions. (**d**) Ala-Phe substrate binding pose, representative cluster frame of 1 μs molecular dynamics (MD) simulation from 7NQK structure with added ligand, for setup details see Materials and methods. Purple dotted lines represent salt-bridge contacts, orange dotted lines other polar contacts. (**e**) ExxER motif salt-bridge cluster, representative cluster frame of 1 μs MD simulation from 7NQK structure.

The online version of this article includes the following figure supplement(s) for figure 1:

**Figure supplement 1.** 1 μs-long molecular dynamics (MD) simulations starting from CHARMM-GUI-embedded and equilibrated PepT2 structures.

**Figure supplement 2.** Metadynamics to derive potential occluded states.

**Figure supplement 3.** Inter-residue heavy atom (H87: NE2, S321: OG, R206: CZ, D342: CG, K64: NZ, D317: CG, D170: CG, K642: NZ) distances for several possible gating interactions.

## Results

### Unbiased MD identifies extra and intracellular gate opening triggers

We began our computational investigation by embedding PepT2 structures (using the sequence of the rat homologue) in the OF (*Parker et al., 2021*), IF (alphafold prediction, *Jumper et al., 2021*), and inwards-facing partially occluded (*Killer et al., 2021*) conformations in 3:1 POPE:POPG membranes (*Figure 1—figure supplement 1*, details in Materials and methods). While we were able to obtain stable wide-open OF and IF simulation boxes, the OCC state required further attention as the MD simulations from the inwards-facing, partially-occluded structure showed embedding artifacts, including extracellular-gate instability and intracellular gate hydrophobic collapse (*Figure 1—figure supplement 1a*). We, therefore, opted to derive an OCC state using metadynamics simulations in 5 replicates (*Figure 1—figure supplement 2*, further details in Materials and methods), using stability in unbiased MD to select the best among the obtained candidates. The OCC state thus developed is validated by the further work in this paper, showing it to be a stable conformational basin that is functionally occluded in that it can open both towards OF and IF in different protonation state conditions (*Figures 2–4*).

Equipped with models of the protein conformations required for PepT2 alternating access (OF, OCC, and IF), we ran triplicate sets of 1 μs-long MD simulations in a range of conditions. To decide which conditions to probe apart from the apo, standard protonation states as obtained above, we investigated the extent to which the H87–S321, R206–D342, K64–D317, and D170–K642 interactions (see *Figure 1b, c*) are formed in the OF (closed intracellular gate) and IF (closed extracellular gate) conformational states (*Figure 1—figure supplement 3*). In the IF state, we found that H87–S321 and D342–R206 are consistently interacting, whereas the K64–D317 interaction, while formed in ≈72% of MD frames, is unstable and of a transient nature and, therefore, unlikely to contribute much to extracellular gate stability. The D170–K642 salt bridge, in turn, is only formed in ≈22% of OF-state frames, thus likely not substantially adding to the stability of the intracellular gate. We, therefore, decided to mainly focus on probing the H87–S321 and D342–R206 interactions with respect to control of the extracellular gate. Since no salt bridges or other specific interactions involving protonatable residues seem to demarcate the intracellular gate, we decided to focus on ExxER motif glutamates (E53 and E56) and E622 for their effects on intracellular gate opening.

Guided thus in our choice of which residues to investigate, we probed whether the OCC state opens spontaneously to OF or IF states in a range of different protonation-state and mutation conditions (as assessed by projection of unbiased MD runs onto intuitive collective variables, or CVs, defined as the centre-of-mass distance between the tips and bases of the N- and C-terminal bundles, respectively, see *Figure 2a*). We found that the extracellular gate remains stably closed in triplicates of 1 μs-long MD when H87 or D342 are protonated individually, but the OCC state can open spontaneously on the simulated time scale to an OF conformation when both are protonated simultaneously (*Figure 2b*; *Figure 2—figure supplement 1a* for plots of the opposite gate in the same trajectories, showing how flexibility of the intra- and extracellular gates is anti-correlated). A comparable effect is found in the presence of the physiological peptide substrate L-Ala–L-Phe (*Figure 2b*, panels 5–6). We have also tested further combinations of mutations and protonation state changes relating to the putative extracellular gating interactions (D317 protonation and mutations of R206, S321, D342 to alanine, with and without H87 protonation) as well as some control mutants which we did not expected to have an effect (I135L, T202A, Q340A, and the salt bridge-swapped mutant R206D & D342R, combined with H87 protonation). Across our 48 * 1 μs unbiased MD runs collated in *Figure 2—figure supplement 2*, we observed three full extracellular-gate opening events, in conditions where H87 was protonated and the D342–R206 salt bridge was also disrupted either by D342 protonation or mutation to alanine (D342A). We also saw one partial opening event when in addition to H87 protonation we also mutated S321 to alanine (S321A). The data thus suggests that for spontaneous extracellular-gate opening to occur on this time scale in unbiased MD, disruption of the OCC-state H87 interaction network is essential, and the D342 salt bridge appears to make an additive contribution towards extracellular-gate stability (though this is not a strict correlation, as illustrated by the S321A partial opening event). The intracellular gate, by contrast, is more flexible than the extracellular gate even in the apo, standard protonation state; however, following either protonation of the conserved E53 and E622 residues or the insertion of Ala-Phe substrate, the intracellular gate becomes more flexible and can spontaneously open

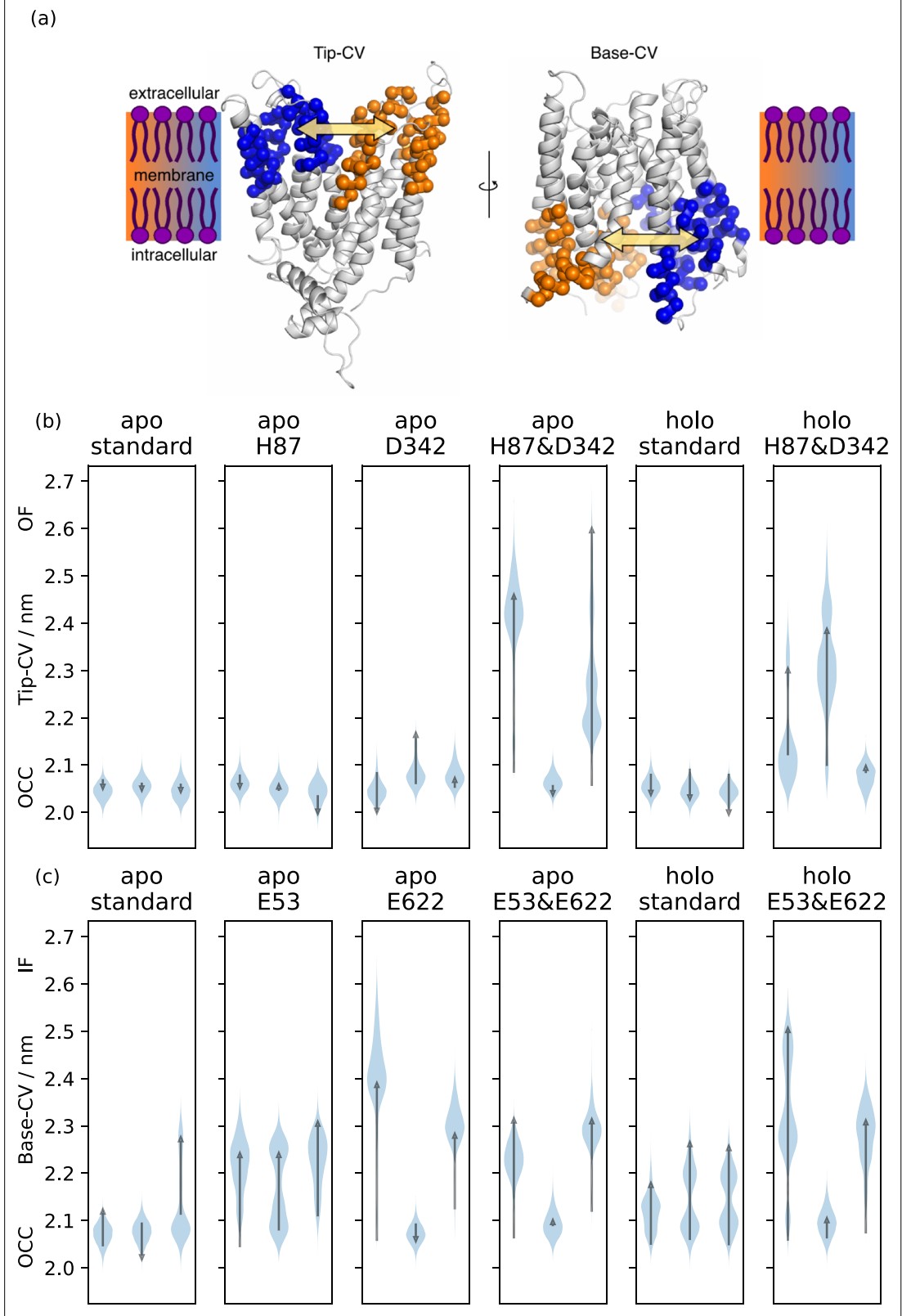

**Figure 2.** Collective variable analysis. (**a**) Illustration of the collective variables (CVs) used to quantify extra-and intracellular gate opening, consisting of inter-bundle centre-of-mass distances between the helical tips (top 11 residues) and bases (bottom 11 residues). (**b**) + (**c**) Triplicate 1 μs-molecular dynamics (MD) simulations starting from occluded (OCC), showing the effects of different protonation and substrate binding states, projected onto the (**b**) Tip-CV and (**c**) Base-CV, respectively. Violin plots are trajectory histograms, arrows link the CV values of the first and last frames.

*Figure 2 continued on next page*

*Figure 2 continued*

The online version of this article includes the following figure supplement(s) for figure 2:

**Figure supplement 1.** Effects of protonation and substrate on vanilla MD simulations of the OCC state.

**Figure supplement 2.** Triplicate 1 μs-molecular dynamics (MD) simulations starting from the occluded (OCC) state, showing the effects of different protonation states and mutations projected onto the tip-collective variables (CV).

**Figure supplement 3.** Illustration of hysteresis effects.

**Figure supplement 4.** 1D-potential of mean forces (PMFs) along the Tip-collective variable (CV) or Base-CV (as indicated), from replica-exchange umbrella sampling (REUS) starting at Morphing Endstates by Modelling Ensembles with iNdependent TOpologies (MEMENTO) intermediates.

**Figure supplement 5.** Illustration of the principal component analysis (PCA)-derived collective variables (CVs) for 2D-replica-exchange umbrella sampling (REUS).

(*Figure 2c*; see *Figure 2—figure supplement 1b* for the corresponding plots of the extracellular gate opening).

Although these unbiased simulations show a large amount of stochasticity and drawing clean conclusions from the data are difficult, we can already appreciate a possible mechanism where protons move down the transporter pore, first engaging H87 and D342 to favour the OF state and then moving to the ExxER motif (E53 and/or E56) and E622 to favour the IF orientation, driving successive conformational changes along the way. The initial unbiased approach taken in this section, therefore, serves to generate hypotheses for testing by a more rigorous investigation of the protonation state-dependent conformational changes. To this end, we set out to employ enhanced sampling simulations for obtaining conformational free energy landscapes in dependence on a range of protonation state and substrate binding conditions.

## 2D-PMFs show proton-dependent driving forces of PepT2 alternating access

To overcome the time-scale limitations of MD simulations and sample important slow degrees of freedom, many enhanced sampling approaches have been developed (*Hénin et al., 2022*) and employed in the computational study of membrane proteins (*Harpole and Delemotte, 2018*). An important class of methods that includes (among others) the popular techniques of steered MD (SMD) (*Izrailev et al., 1999*), umbrella sampling (*Torrie and Valleau, 1977*), metadynamics (*Barducci et al., 2008*), adaptive biasing force (ABF) (*Darve et al., 2008*), and the accelerated weight histogram method (AWH) (*Lindahl et al., 2014*) uses a small number of collective variables (CVs) along which to bias the simulation, thus improving exploration of important regions of phase space if the CV includes the relevant slow degrees of freedom (DOFs). If the CV is not optimal, problems can manifest in the form of hysteresis (starting-state dependence) when moving between known end-states (*Lichtinger and Biggin, 2023*). This is the case for the PepT2 OCC↔OF transition with the simple tip-CV illustrated in *Figure 2a*. Using either metadynamics or SMD with replica-exchange umbrella sampling (REUS), the end-state from which the simulations were started is always favoured in the resulting potential of mean force (PMF), with the hysteresis effect totalling ≈40 kcal mol⁻¹ for each method (*Figure 2—figure supplement 3*).

We have recently developed a strategy to overcome such hysteresis issues in conformational sampling which we call MEMENTO (Morphing Endstates by Modelling Ensembles with iNdependent TOpologies), (*Lichtinger and Biggin, 2023*), and have implemented it as the freely available and documented PyMEMENTO package. MEMENTO generates paths between known end-states by coordinate morphing followed by fixing the geometries of un-physical morphed intermediates. Since these paths by definition connect the correct end-states (unlike biased MD methods like SMD, where not reaching the target state in slow orthogonal DOFs is a common source of hysteresis), they can drastically reduce hysteresis in enhanced sampling compared to SMD as a path generation method. We have shown this for several validation cases, including a large-scale conformational change in the bacterial membrane transporter LeuT. After running the initial 1D-REUS from MEMENTO replicates along a simple CV guess, we can use the generated MD data to iteratively improve CVs using principal component analysis (PCA), thereby capturing slow motions from long end-state sampling that propagates through MEMENTO as differences between replicates.

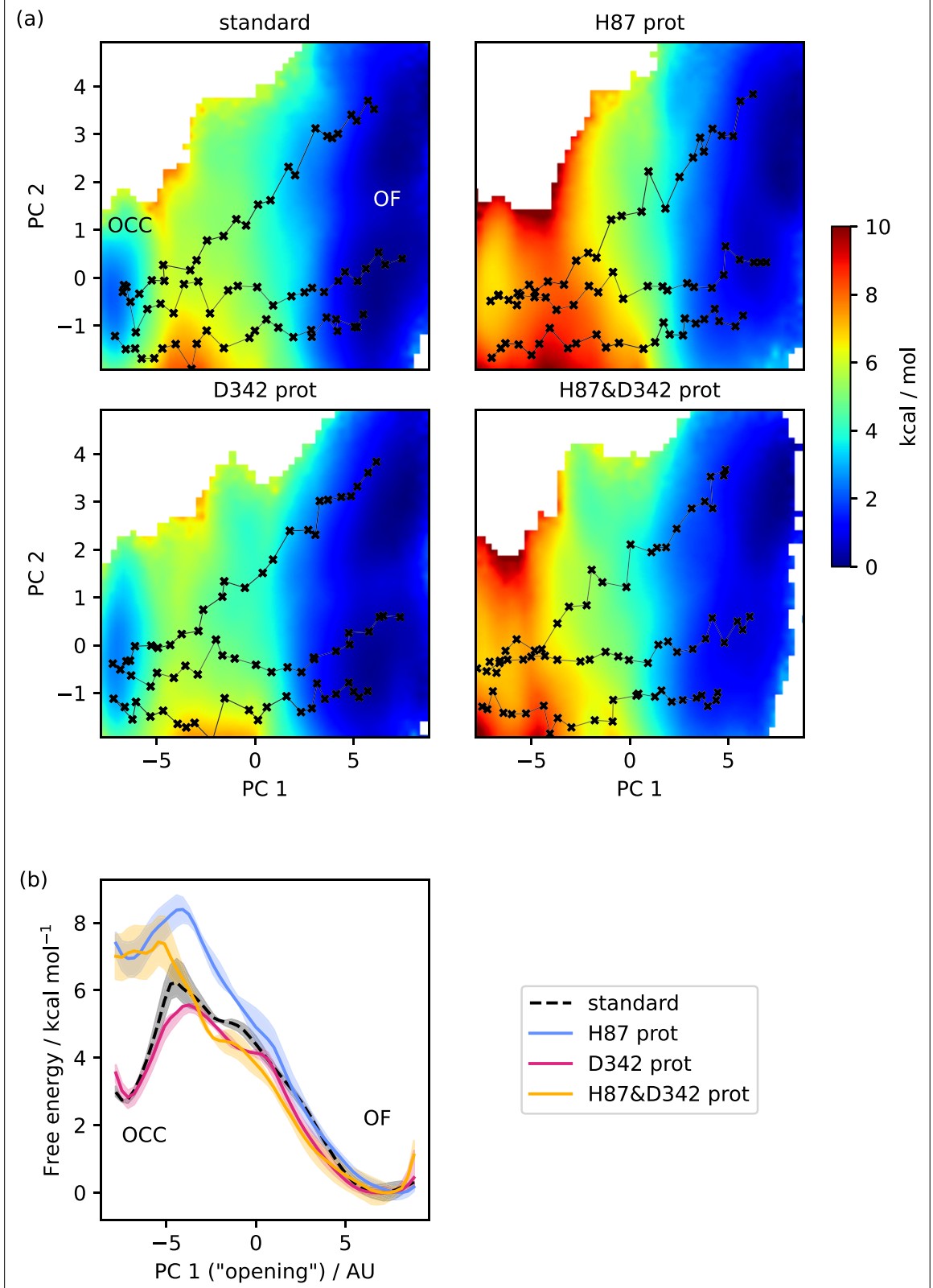

**Figure 3.** Free energy surfaces for the OCC-OF transition. (**a**) 2D-PMFs from replica-exchange umbrella sampling (REUS) starting with Morphing Endstates by Modelling Ensembles with iNdependent TOpologies (MEMENTO) paths, in different protonation states of candidate extracellular gating residues. (**b**) Projection of the 2D-PMFs in part a onto PC 1 using Boltzmann reweighting. Shaded areas indicate convergence errors as the range of PMF values for a given CV value obtained with the first 40%, the last 40%, and 100% of sampling included (after alignment to the 100% curve). H87 and

*Figure 3 continued on next page*

*Figure 3 continued*

D342 form an additive extracellular gate, where H87 protonation changes the relative occluded (OCC)–outwards-facing (OF) state energies as well as the transition barrier, while D342 protonation only contributes in the transition region. Note that the individual PMFs are only determined by our REUS approach up to additive constants, and are shown aligned here at the OF state for convenience of comparison.

The online version of this article includes the following figure supplement(s) for figure 3:

**Figure supplement 1.** 2D-PMFs of the OCC↔OF transition from replica-exchange umbrella sampling (REUS) with Morphing Endstates by Modelling Ensembles with iNdependent TOpologies (MEMENTO) paths in additional protonation state and mutation conditions.

**Figure supplement 2.** All available OCC↔OF 2D-PMFs, projected onto the first collective variable (CV) (PC 1).

**Figure supplement 3.** Interaction plots of the 2D-PMF trajectory data (high force constant windows only), calculated as frequencies of finding inter-residue heavy-atom distances smaller than 0.35 nm, shown as a line for the average across three replicates with shaded standard deviations.

**Figure supplement 4.** Convergence plots of all OCC↔OF 2D-PMFs, shown as projections onto PC 1 including successively (increasing saturation) more data points, starting from the first frame (green) or from the last frame in reverse (orange).

**Figure supplement 5.** 2D-replica-exchange umbrella sampling (REUS) histograms for the OCC↔OF standard protonation state 2D-PMF, drawn as contour lines at 30% of the maximal histogram height for each window (coloured by window, from purple at occluded (OCC) to yellow at outwards-facing (OF)).

Here, we ran triplicates of MEMENTO for the OCC↔OF (standard protonation states and H87&D342 protonated) and OCC↔IF (standard protonation states and E53 protonated) conformational changes, followed initially by 1D-REUS along the tip-CV (*Figure 2—figure supplement 4*). The results are much more consistent than SMD or metadynamics along the same CV, and the shapes of the PMFs fit well with the trends we previously observed in unbiased MD from the OCC state. Since, however, significant differences between replicates remained, we used principal component analysis (PCA, see Materials and methods for details) on the sampling collected of the standard protonation state transitions to derive sets of 2-dimensional CVs (*Figure 2—figure supplement 5* and *Videos 1–4*) that capture the main gate-opening motions in the first PC, and the direction along which the differing replicates can be best separated out as the second PC (these correspond to cleft sliding and twisting motions).

Equipped with these CVs, we first studied the protonation-state dependence of the OCC↔OF transition. As *Figure 3* shows (2D-PMFs in part a, projections onto PC 1 in part b), the OCC state in standard protonation states form a basin that is metastable with respect to OF (lying ≈3 kcal mol$^{-1}$ higher than OF, separated by a barrier of ≈3 kcal mol$^{-1}$). Protonation of H87 still leads to a metastable OCC basin, although it is raised by ≈4 kcal mol$^{-1}$ and the barrier is decreased to ≈1.5 kcal mol$^{-1}$. Protonating D342, in turn, does not affect the relative free energies of the OCC and OF states, but does lower the transition barrier by ≈1 kcal mol$^{-1}$. These effects are additive, so that protonation of both H87 and D342 abolishes the metastable OCC state-an observation which agrees with the ability of the OCC state thus protonated to spontaneously transition to OF in unbiased MD (see the Discussion for a comparison with the results obtained by *Parker et al., 2017* on PepT$_{So}$ on this point).

We have also computed 2D-PMFs in further protonation-state and mutation conditions to gain a better understanding of how the H87 and D342–R206 interaction networks control the extracellular gate (*Figure 3—figure supplement 1* for the 2D-PMFs, all 1D-reprojections are shown for reference in *Figure 3—figure supplement 2*). From the data presented thus far, it is not clear whether the effect of H87 pronation on the OCC → OF transition is due merely to the loss of hydrogen-bond interactions with S321, or whether the introduction of a positive charge in this location makes a significant mechanistic contribution. To address this question, instead of protonating H87 we mutated it to alanine (H87A). In the resulting PMF, the OCC state is raised less with respect to OF compared to the protonated version, and the transition barrier increases to ≈5 kcal mol$^{-1}$, suggesting that there exists an interaction made by positively charged H87 that becomes particularly relevant in the transition region. Further analysis of the H87 interaction networks in our 2D-REUS trajectories (*Figure 3—figure supplement 3a*) reveals that when H87 is protonated, the interaction with S321 is substituted by an interaction with D317 that is strongest in the transition region. Interestingly, this suggests an alternative mechanistic role for the essential D317, which-as discussed above-we have not found forming the structurally observed salt bridge with K64 consistently in our simulations.

An equivalent investigation of the D342A mutation results in a PMF that shows both a decrease in the transition barrier, and-as opposed to D342 protonation-also raises the OCC state in energy with respect to OF. This may be explained by the fact that protonated D342 can still hydrogen-bond with

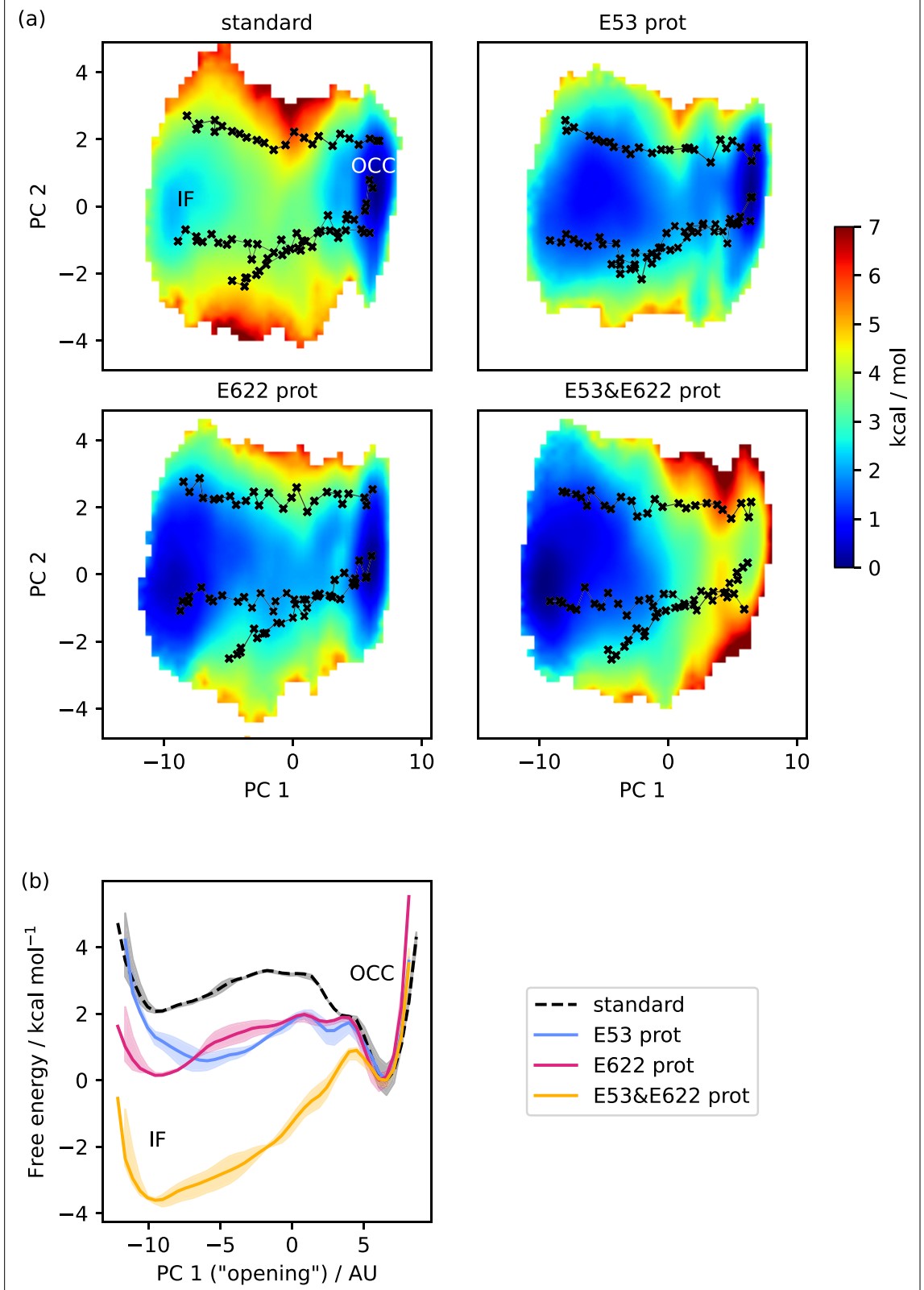

**Figure 4.** Free energy surfaces for the OCC-IF transition. (**a**) 2D-PMFs from replica-exchange umbrella sampling (REUS) starting with Morphing Endstates by Modelling Ensembles with iNdependent TOpologies (MEMENTO) paths, in different protonation states of candidate intracellular gate-controlling residues. (**b**) Projection of the 2D-PMFs in part a onto PC 1 using Boltzmann reweighting. Shaded areas indicate convergence errors as the range of PMF values for a given collective variable (CV) value obtained with the first 40%, the last 40%, and 100% of sampling included (after alignment

*Figure 4 continued on next page*

*Figure 4 continued*

to the 100% curve). E53 and E622 protonation have additive and approximately equal effects on driving the OCC→IF transition. Note that the individual PMFs are only determined by our REUS approach up to additive constants, and are shown aligned here at the IF state for convenience of comparison.

The online version of this article includes the following figure supplement(s) for figure 4:

**Figure supplement 1.** 2D-PMFs of the OCC↔IF transition from replica-exchange umbrella sampling (REUS) with Morphing Endstates by Modelling Ensembles with iNdependent TOpologies (MEMENTO) paths in additional substrate-bound protonation state conditions.

**Figure supplement 2.** All available OCC↔IF 2D-PMFs, projected onto the first collective variable (CV) (PC 1).

**Figure supplement 3.** Convergence plots of all OCC↔IF 2D-PMFs, shown as projections onto PC 1 including successively (increasing saturation) more data points starting from the first frame (green) or from the last frame in reverse (orange).

R206, so although the interaction is less prominent it presumably still contributes somewhat to OCC state stability (*Figure 3—figure supplement 3b*). As a control, we also show that although the protonation of E53/E56 can affect the relative OCC vs OF free energies, there is no lowering of the transition barrier (while they do have an effect on the transition barrier separating OCC from IF, as discussed below). Convergence analysis and representative 2D-REUS histograms for our OCC ↔ OF PMFs can be found in *Figure 3—figure supplement 4 and 5*.

We next employed an equivalent approach to investigate the OCC↔IF transition. As can be seen from *Figure 4*, in standard protonation states, the OCC state forms a well-defined basin, connecting to a broader and shallower IF basin raised ≈2 kcal mol$^{-1}$ over OCC via a barrier of just over ≈3 kcal mol$^{-1}$. When either E53 or E622 are protonated, IF drops to a similar free energy level as OCC and the transition barrier lowers by ≈1 kcal mol$^{-1}$. These effects are additive, so that when both E53 and E622 are protonated, the IF state is favoured by ≈3.5 kcal mol$^{-1}$, and is accessible from OCC via a barrier of only ≈1 kcal mol$^{-1}$. As for the OCC↔OF transitions, these results explain the behaviour we had previously observed in the unbiased MD of *Figure 2c*. The stochastic partial intracellular gate opening seen with those runs can be rationalised through the lower transition barrier from OCC to IF compared to the transition to OF in our PMFs, together with the broad and flat shape of the IF-state basin. Additionally, we have also computed all the equivalents of these PMFs in the presence of Ala-Phe substrate (*Figure 4—figure supplement 1*), which we will discuss in the section below. All projections onto PC 1 are shown in *Figure 4—figure supplement 2*, and convergence analysis is provided in *Figure 4—figure supplement 3*. Taken together, the PepT2 apo 2D-PMFs provide a detailed view of the way the alternating access cycle is driven by proton movement from the extracellular to the intracellular side of the transporter that fits well with its biological function of using two protons from the extracellular medium to energise cycling from OF to IF, and back spontaneously once the protons have left (see the Discussion below).

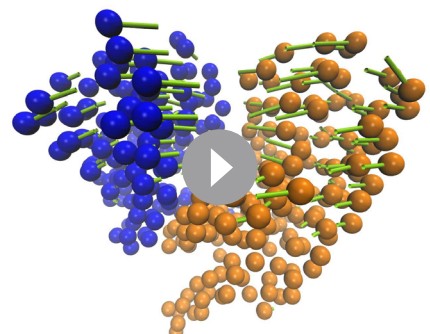

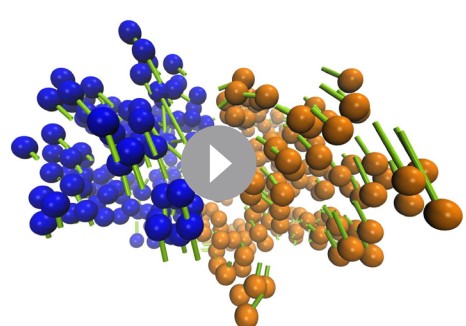

**Video 1.** First Principal Component (PC) of occluded (OCC) to outwards-facing (OF) state.
https://elifesciences.org/articles/96507/figures#video1

**Video 2.** Second Principal Component (PC) of occluded (OCC) to outwards-facing (OF) state.
https://elifesciences.org/articles/96507/figures#video2

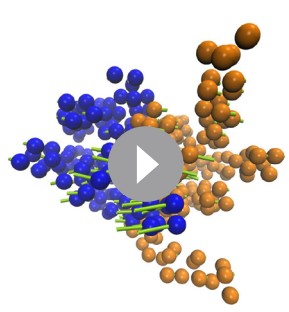

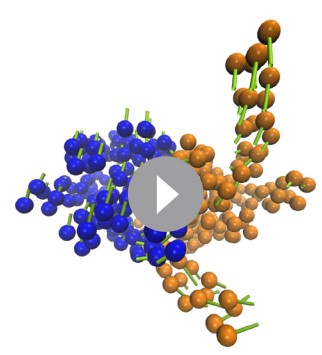

**Video 3.** First Principal Component (PC) of occluded (OCC) to inwards-facing (IF) state.
https://elifesciences.org/articles/96507/figures#video3

**Video 4.** Second Principal Component (PC) of occluded (OCC) to inwards-facing (IF) state.
https://elifesciences.org/articles/96507/figures#video4

## Substrate coupling of alternating access includes several distinct mechanisms

Given the evidence presented so far, which provides a plausible model for how protons drive alternating access based just on an investigation of the apo states, it remains unclear how coupling of proton transport to the substrate is achieved-that is, why the transporter cannot be driven by protons without the presence of substrate (and would thus just leak protons across the membrane). To investigate the mechanism underpinning peptide–proton coupling, we constructed simulation boxes that included a bound Ala-Phe substrate molecule (see Materials and methods for details). We then calculated the Ala-Phe affinity using ABFE simulations in different protein states (*Table 1*). First, we observed that the affinity is similar in the OF and IF states, indicating that the binding of substrate alone does not thermodynamically drive the transporter from OF to IF. We did find, however, that protonation of E622 (i.e. the salt bridge parter of the substate N-terminus) significantly decreases substrate affinity. Given that protonation of E622 also favours the OCC→IF transition, we suggest a dual function of E622 protonation that includes both stabilising the IF state with respect to OCC and facilitating substrate release from holo IF.

To explore what effect the substrate has on the PepT2 conformational landscape, we repeated a set of our 2D-PMFs in the presence of substrate (*Figure 5*). For both the OCC↔OF and OCC↔IF transitions, the 2D-PMFs have similar shapes in the apo and holo states. For OCC↔OF, however, we found an increased width of the OCC state basin in the direction of PC 2, which-as is evident after projection onto PC 1-stabilises OCC with respect to OF by ≈1 kcal mol$^{-1}$ as an effect of increased OCC flexibility in the orthogonal DOF. We find a similar stabilisation when E53 or E56 are protonated, but not when both H87 and D342 are protonated (*Figure 3—figure supplement 1 and 2*). This indicates that the presence of substrate in conformations approaching the OCC state from OF may trigger proton movement further down into the transporter-driven by entropic gains from increased flexibility in orthogonal DOFs.

**Table 1.** Results of absolute binding free energy (ABFE) calculations, showing that the affinity of Ala-Phe substrate does not depend much on the conformational state (outwards-facing, OF vs inwards-facing, IF), but is significantly decreased on E622 protonation.

| Protein state | AF dipeptide affinity/kcal mol$^{-1}$ |
| --- | --- |
| OF | 8.0 ± 0.3 |
| IF | 7.0 ± 0.4 |
| IF & E622 prot | 2.9 ± 0.2 |

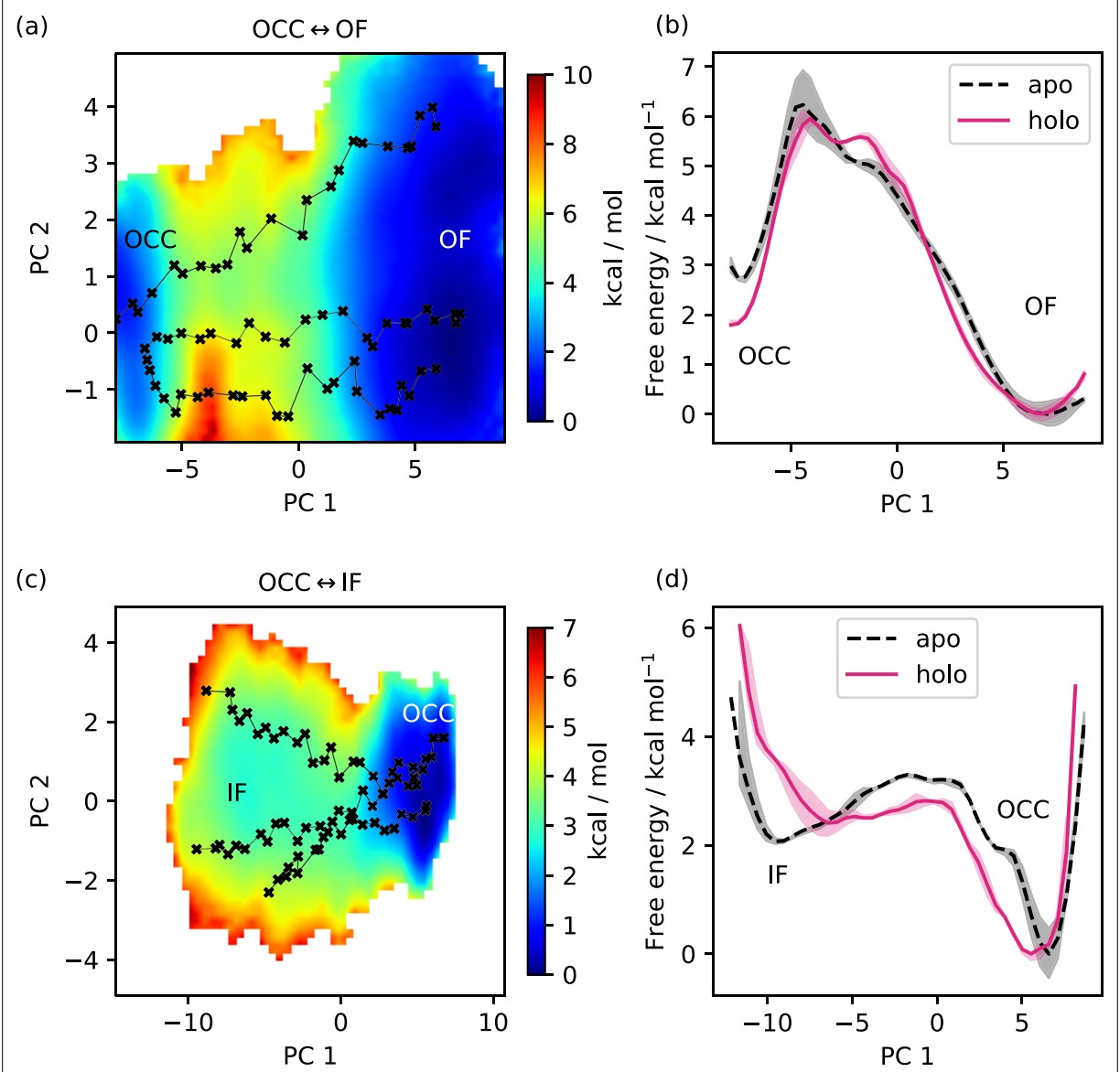

**Figure 5.** Dependence of free energy surface on presence or absence of substrate. (**a**) 2D-PMF for the OCC↔OF transition from replica-exchange umbrella sampling (REUS) starting with Ala-Phe-bound PepT2 Morphing Endstates by Modelling Ensembles with iNdependent TOpologies (MEMENTO) paths. The occluded (OCC) state has an increased basin width in PC 2 (compared to *Figure 3a*), and a transition path shifted in PC 2. (**b**) Projection of the PMF from panel a onto PC 1, showing how in holo PepT2, the OCC state is stabilised by ≈1 kcal mol⁻¹. Shaded areas indicate convergence errors as the range of PMF values for a given collective variable (CV) value obtained with the first 40 %, the last 40%, and 100% of sampling included (after alignment to the 100% curve). Note that the individual PMFs are only determined by our REUS approach up to additive constants, and are shown aligned here at the OF state for convenience of comparison. (**c**) 2D-PMF for the OCC↔IF transition from REUS starting with Ala-Phe-bound PepT2 MEMENTO paths. The structure of the inwards-facing (IF) plateau is not significantly affected, but OCC is more flexible in PC 1. (**d**) Projection of the PMF from panel c onto PC 1, showing how in holo PepT2, the OCC state has a broader basin, corresponding to intracellular-gate flexibility. Convergence error and alignment of PMFs are shown in panel b.

The OCC↔IF PMF also presents a broader OCC basin in the presence of substrate, this time in the direction of PC 1 (consistent with the higher OCC-state flexibility directed towards IF observed in unbiased MD, see *Figure 2c*). In the projection onto PC 1, this manifests as a broader free energy well and a lower barrier towards IF, even if the relative energies of the basins are not significantly affected. This effect is similarly apparent when E53 is protonated, but not with E622 protonation, which instead leads to a raised transition barrier ( *Figure 4—figure supplement 1 and 2*). We reason that this is due to a more flexible substrate orientation (disengaging the N-terminus) when E622 is neutral. While,

taking these observations together, substrate binding does lead to an OCC→IF bias, it also seems unlikely that E622 is protonated at the moment when the OCC→IF transition happens in the holo transporter. This may suggest a further intermediate protonation step that our simulations have not captured (see our Discussion below).

As noted above, another possibility for the substrate to engage with the transport cycle is found in R57 interacting both with the substrate C-terminus and with the ExxER glutamates, protonation of which drives the transporter towards the IF state, as our PMFs demonstrate. A natural hypothesis then is that substrate binding-which engages R57-loosens the R57 interaction with E53 thus enabling the protonation of those residues and progress along the alternating access cycle. To test this hypothesis, we conducted triplicate constant-pH simulations (CpHMD) (*Swails et al., 2014*) to probe the E53 and E56 pKa values in all combinations of OCC/OF and apo/holo conditions (*Figure 6a*). Concerning the E53 pKa value, we see a potential response to substrate binding in the OF state (though the error bars calculated from the triplicate standard deviations overlap) but not in OCC. On the other hand, we do see a raising of the E56 pKa beyond error in holo OF or OCC states compared to apo, amounting to ≈0.6 log units in both cases. As shown in *Figure 6—figure supplement 1 and 2*, the pKa values estimated for successive data chunks across the CpHMD trajectories vary significantly with simulation time in a complex superposition of timescales, and with a dynamic range larger than the replicate error bars. As an alternative to the replicate-based representation in *Figure 6*, we have, therefore, also analysed the pooled data for each condition as histograms of pKa values estimated from short chunks of our simulations (*Figure 6—figure supplement 3*). From this, we recover the same effect of substrate binding on the E56 pKa in the OF and OCC states, as well as a potential effect on the E53 pKa in the OF state only. Since the shift in E56 pKa was more robust across conformational states, we focus on this residue in the following validation and discussion, although we note that if there was in fact a significant raising of the E53 pKa as well, this would further strengthen our conclusions about substrate coupling to E53 and/or E56 protonation.

It is important to recognise here that the hybrid-solvent CpHMD method as implemented in AMBER is not rigorous for membrane proteins, since the membrane is not taken into account while evaluating proposed protonation state changes in implicit solvent. On the other hand, we were not able in initial trials to obtain sufficient transition counts to converge an alternative explicit-solvent CpHMD method as implemented in GROMACS (*Aho et al., 2022*). To validate our results, we, therefore, also constructed a thermodynamic cycle of substrate binding and E56 protonation in the OF state by including data from separate ABFE calculations as connecting thermodynamic legs (*Figure 6b*). The ABFE results show a complementary effect of the E56 protonation state on substrate affinity, closing the thermodynamic cycle, and validating our CpHMD simulations using an orthogonal MD technique. We conclude that substrate binding does indeed facilitate protonation of E56 (whence, we stipulate, the proton moves to E53, which is situated in close proximity). If the neglected membrane environment in the hybrid-solvent CpHMD did produce significant artifacts in the pKa values, then it would appear that there is error cancellation when assessing the impact of substrate binding as a difference of pKa values in the apo and holo conditions.

It should be noted that-as throughout this study, see the discussion below-in studying the coupling between substrate binding and protonation-state changes at the E53 and E56 we have not made the substrate C-terminus protonatable. Since, in order to induce E56 protonation, the substrate C-terminus needs to engage R57 in a salt bridge, its pKa is likely to be low, rendering the assumption reasonable for those substrate conformations. However, it is possible that the system could also adopt states in which the ExxER motif salt-bridge network is stable in a way similar to the apo condition while the substrate gets protonated when oriented away from R57. If such conformations contribute significantly to this semi-grand canonical ensemble, the E53 and E56 (of the ExxER motif) pKa values estimated without taking them into account may exhibit some bias. By undersampling more apo-like conformations in the holo state in this way, it is possible that the calculations presented here overestimate the substrate-induced pKa shift of E56, although the direction of change would be expected to be the same, because the substrate can still engage R57 when it is deprotonated (we speculate that the histograms in *Figure 6—figure supplement 3* for the holo state may become bi-modal in this case). While the possibility would need to be taken into account for a more rigorous quantitative estimate of the E53 and E56pKa value shifts, the pKa calculations would become much harder to converge since the slow degree of freedom of substrate re-orientation would need to equilibrate to

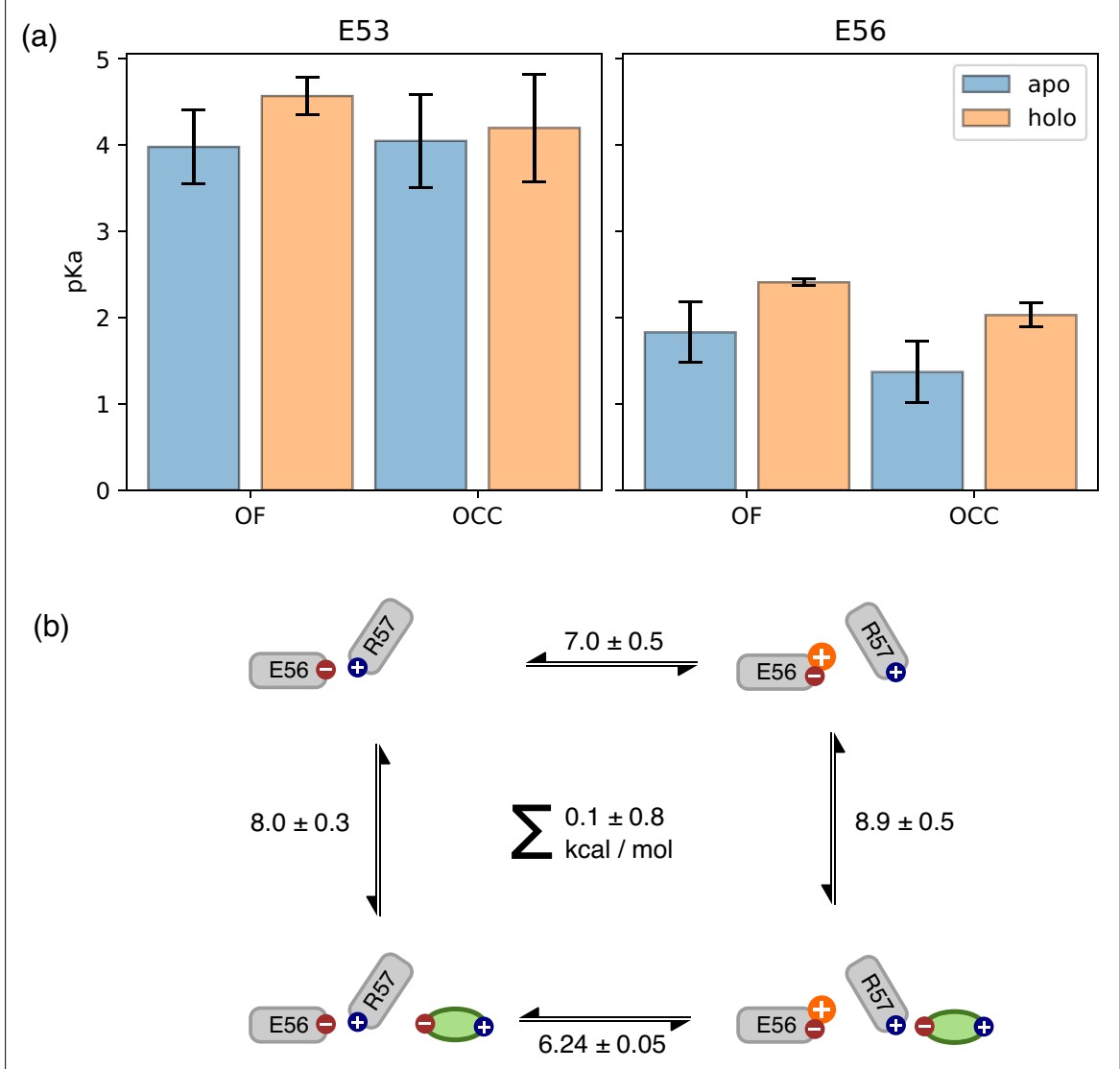

**Figure 6.** Use of constant pH simulations to investigate influence of substrate on pKa values. (**a**) E53 and E56 pKa values from constant-pH molecular dynamics (MD) simulations, in the apo and holo as well as the outwards-facing (OF) and occluded (OCC) states, estimated as mean ± standard deviation from triplicate runs (using the full simulation data for fitting the titration curves). The presence of substrate raises the E56 pKa in either conformational state, while some effect on the E53 pKa may also exist in the OF state. (**b**) Illustration of a themodynamic cycle of E56 protonation and Ala-Phe binding, with edges filled in via constant pH simulations (CpHMD) (converted into kcal mol$^{-1}$ at pH 7) for the top and bottom transitions, and absolute binding free energy (ABFE) for the left and right edges. Notably, ABFE displays a response of Ala-Phe affinity to E56 is consistent with the CpHMD results, and the cycle closes very well. The error in the cycle closure residual is estimated as a square root of the sum of squared standard deviations of the individual edges.

The online version of this article includes the following figure supplement(s) for figure 6:

**Figure supplement 1.** E53 pKa values estimated for (separate) successive chunks of 80 ns (10 ns per pH window) of constant pH simulations (CpHMD) via fitting the Hill equation.

**Figure supplement 2.** E56 pKa values estimated for (separate) successive chunks of 80 ns (10 ns per pH window) of constant pH simulations (CpHMD) via fitting the Hill equation.

**Figure supplement 3.** Histograms of the pKa values estimated from chunks of 80 ns (10 ns per pH window) of constant pH simulations (CpHMD), pooled for all trajectories of a given condition.

the protonation-state changes (which happen fast since they are treated as Monte-Carlo moves in an implicit solvent at regular intervals). Here, we content ourselves with the more qualitative observation that an appropriately positioned substrate in the canonical, structurally observed binding pose facilitates protonation of the ExxER glutamates.

### Validation in cell-based transport assays

To experimentally validate the results of an MD investigation, the first step is to probe the importance of the key implicated residues for protein function. We note that the literature already contains ample data to show that E53 (*Solcan et al., 2012*; *Doki et al., 2013*; *Sun et al., 2014*; *Jørgensen et al., 2015*; *Parker et al., 2017*), E56 (*Solcan et al., 2012*; *Jørgensen et al., 2015*; *Sun et al., 2014*; *Zhao et al., 2014*; *Parker et al., 2017*), R57 (or the equivalent arginine) (*Solcan et al., 2012*; *Guettou et al., 2013*; *Doki et al., 2013*; *Jørgensen et al., 2015*; *Lyons et al., 2014*; *Parker and Newstead, 2014*; *Sun et al., 2014*; *Parker et al., 2017*; *Martinez Molledo et al., 2018*), H87 (for those homologues which conserve it) (*Fei et al., 1997*; *Chen et al., 2000*; *Newstead et al., 2011*; *Uchiyama et al., 2003*; *Parker et al., 2017*), D317 (or the equivalent glutamate) (*Solcan et al., 2012*; *Doki et al., 2013*; *Lyons et al., 2014*; *Parker et al., 2017*; *Martinez Molledo et al., 2018*; *Shen et al., 2022*), and E622 (*Solcan et al., 2012*; *Guettou et al., 2013*; *Doki et al., 2013*; *Lyons et al., 2014*; *Zhao et al., 2014*; *Parker et al., 2017*; *Minhas and Newstead, 2019*; *Shen et al., 2022*) are important for transport through POTs. Of the residues implicated by our simulations, therefore, only S321 and D342 have not been studied before, and thus serve as predictive validation test cases here.

Using cell-based transport assays (see Materials and methods), we tested the transport activity of rat PepT2 WT and several mutants: H87A (as a positive control known from the literature), I135L (as a negative control, without any expected effect), as well as the mutants of interest D342A and S321A (*Figure 7*, and *Figure 7—figure supplement 1* for loading control and membrane localisation micrographs). We note that all our mutants expressed slightly less compared to the WT at the same amount of transfected DNA (0.8 µg), but more than WT at a reduced transfection DNA level (0.5 µg) (*Figure 7b*). To control for this difference in expression levels, we took WT (0.5 µg), which transports ≈20% less than WT (0.8 µg), as a lower bound for the WT transport activity and as the point of comparison for statistical tests. We found that all mutants of residues predicted to be involved in transport displayed significantly reduced transport activity (p-values: $2.2 \times 10^{-5}$ for H87A, $1.6 \times 10^{-4}$ for D342A, $6.9 \times 10^{-5}$ for S321A, while I135L is indistinguishable from WT at *P*=0.79). We also note that D342A, although its activity is significantly reduced, still transports more than H87A (p = $3.9 \times 10^{-7}$). This fits well with our 2D-PMF results, where H87 protonation does more than D342 protonation to stabilise OF with respect to OCC.

## Discussion

Integrating the results from extensive sampling across several MD methods, covering all stages of the PepT2 alternating access cycle, we are now in a position to propose a detailed molecular mechanism of the complete transport cycle, including accounts both of proton coupling to conformational changes and of substrate–proton coupling (*Figure 8*). Starting at the apo OCC state without any protons bound, we find in our 2D-PMFs that proton binding to H87 and D342 stabilises OF with respect to OCC (with H87 being the major contributor). Given that H87 and D342 are accessible from the acidic extracellular bulk, and in light of the transition-region stabilisation from the H87 (protonated)–D317 interaction we have identified, our simulations suggest an interpretation where protonation happens in the OCC state, driving the OCC→OF transition by stabilising the OF state over OCC. However, if the OCC→OF transition is kinetically accessible on experimental timescales without prior protonation events (beyond what our MD was able to sample), it would also be consistent with our data that OCC→OF is spontaneous in standard protonation states, and H87 and (possibly) D342 are merely the initial sites of protonation once OF is reached, providing further stabilisation. [It may seem like the latter model is favoured by the fact that in our PMFs, OCC lies higher than OF, even when neither H87 nor D342 are protonated. We believe, however, that there is a danger of overinterpreting this feature of the PMF. Any combination of effects from forcefields, lipid composition, and the population shifts afforded by transmembrane electrochemical gradients could perturb the conformational equilibria. It would, therefore, not be meaningful to interpret the shape of a single

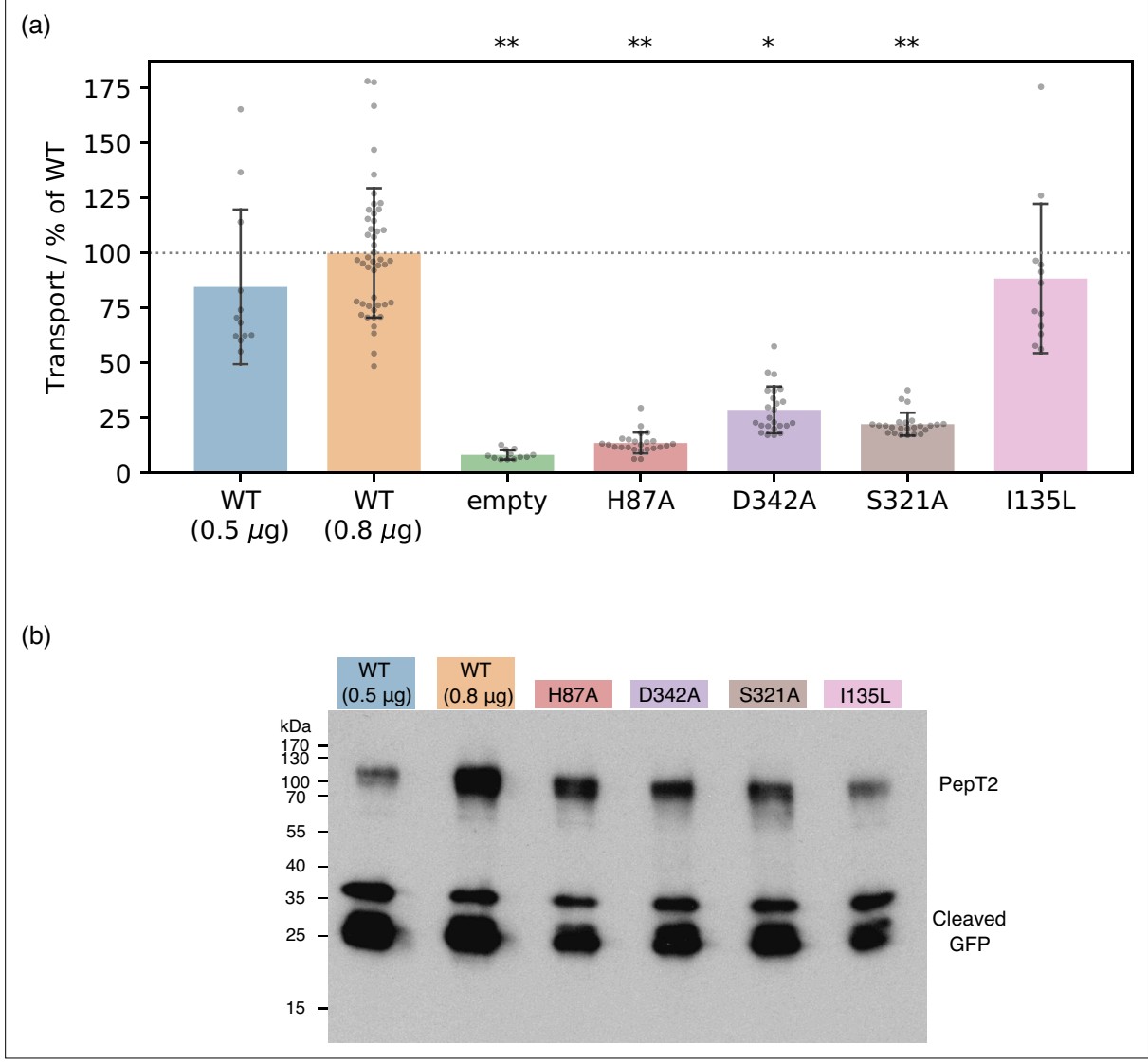

**Figure 7.** Experimental validation of computational predictions. (**a**) Cell-based transport assays for PepT2 wild-type (WT) (transfected with 0.5 μg, n=12, and 0.8 μg, n=46, of DNA per well), empty plasmid vector (n=12) and PepT2 H87A, D342A, S321A (n=24 each) and I135L (n=12) mutants, all transfected with 0.8 μg of DNA. Diagram shows transport as fluorescence in post-assay lysate divided by total protein concentration, normalised to the WT (0.8 μg) mean. Bars are mean values plus minus standard deviation, and swarm plots samples corresponding to individual wells. Single asterisk indicates $p < 10^{-3}$, double asterisks $p < 10^{-4}$ significance levels for difference compared to (weaker transporting, 0.5 μg-transfected) WT, as evaluated using a two-tailed t-test. (**b**) Western-blot showing expression levels of WT and mutant GFP-labelled PepT2, with an anti-GFP primary antibody. All mutants express at levels between the WT transfected with 0.5 μg and 0.8 μg plasmid DNA. Cleaved GFP is also visible at low molecular weight, at levels comparable for WT and mutants.

The online version of this article includes the following source data and figure supplement(s) for figure 7:

**Source data 1.** Raw gel image for *Figure 7b*.

**Figure supplement 1.** Control data for cell-based transport assays.

**Figure supplement 1—source data 1.** Raw gel image for *Figure 7—figure supplement 1*.

---

PMF in this way; only responses of the PMF to protonation-state or substrate-binding changes should be used to inform our view of the conformational cycle, since these are likely to benefit from error cancellation with respect to factors that act on the overall protein conformations (which are conserved between the different conditions in which the PMFs were sampled).] We note in this context the limitation of using non-reactive MD methods for sampling the PepT2 conformational changes. In principle, a multi-dimensional PMF calculated with a reactive MD method where one CV is an explicit

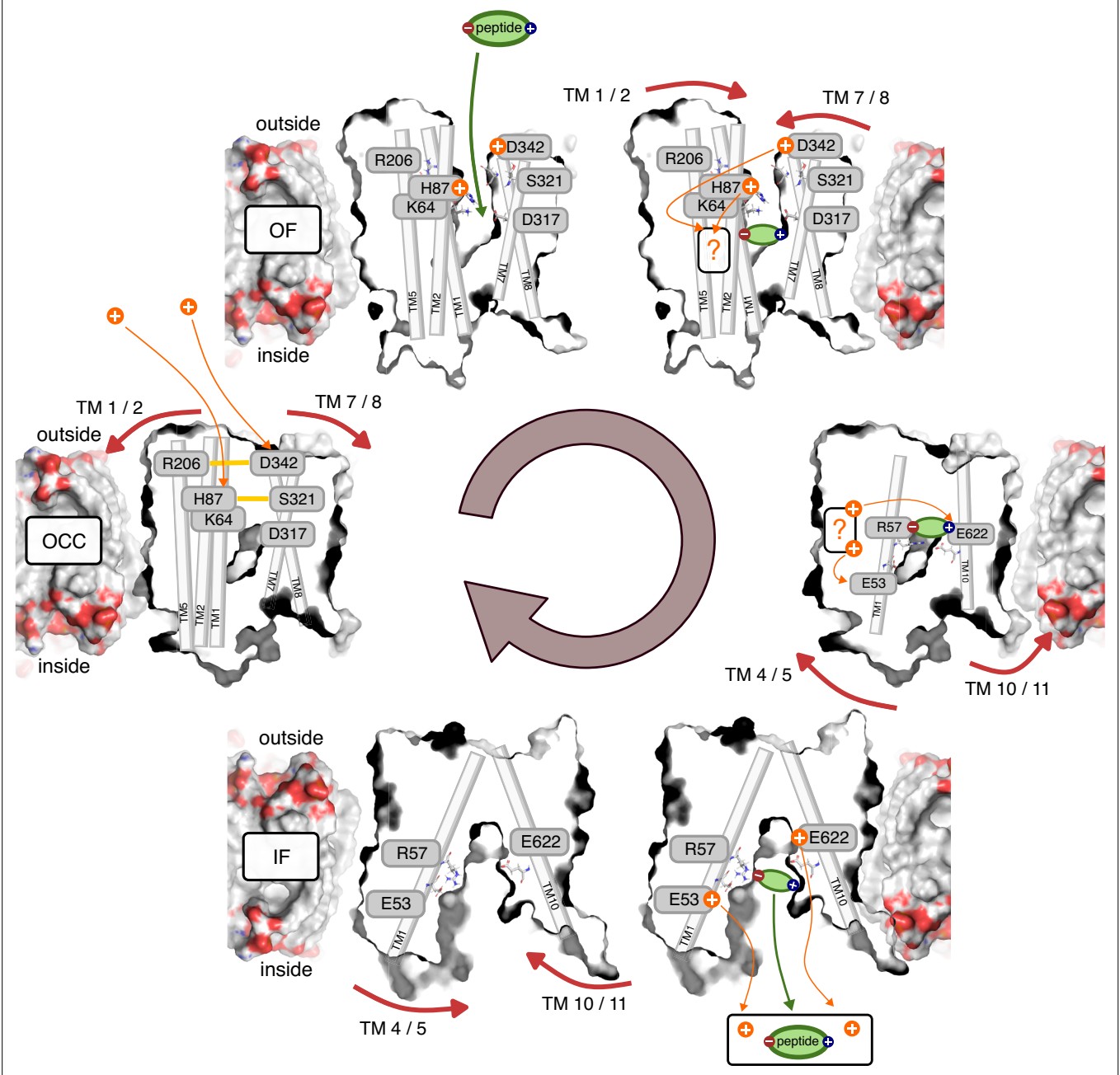

**Figure 8.** Schematic overview of the PepT2 alternating-access transport cycle proposed in this work. Protons located at a question mark indicate a proton-transfer step with an as-of-yet unknown mechanism regarding intermediate residues.

proton movement coordinate could disambiguate between the possible scenarios here. However, we believe that such a fully coupled treatment of proton movement and large-scale conformational changes is not yet computationally feasible, and focussed here on achieving convergence of the conformational sampling in discrete protonation-state combinations. The question of whether proton binding happens in OCC or OF, therefore, warrants further investigation, and indeed the co-existence of several mechanisms may be plausible. Nonetheless, this study contributes important details to a mechanistic understanding of the thermodynamics of proton-coupled alternating access.

Our results with regards to the driving forces behind the OCC→OF motion agree with the work of *Parker et al., 2017*, who found spontaneous opening of PepT$_{Xc}$ towards OF after protonating the equivalent histidine to H87 (PepT2). In transporters which do not conserve the mammalian histidine at

the TM 2 position such as PepT$_{St}$ (*Batista et al., 2019*) and PepT$_{Sh}$ (*Li et al., 2022*), on the other hand, previous simulation studies have implicated protonation of the glutamates equivalent to D317 (PepT2) in the opening of the extracellular gate. Our simulations suggest that the involvement of D317 in the extracellular gating mechanism of PepT2 is by interacting with the protonated H87 to stabilise the transition region for extracellular gate opening. The mechanism of extracellular gating is, therefore, conserved less widely than the overall alternating access mechanism. This point is further highlighted by our results indicating a role of the D342–R206 salt bridge, which is conserved only among mammalian POTs, but not in PepT$_{So}$ and PepT$_{Xc}$ which do have the TM 2 histidine. This may explain why spontaneous opening in unbiased MD was observed by *Parker et al., 2017* for PepT$_{Xc}$ following just H87 protonation, while for PepT2 in this work, protonation of D342 is also required.

Once the transporter is in the OF conformation, the substrate enters the binding site. We have treated this step alchemically in our ABFE simulations, so that the data presented here is agnostic of the orientation of the entering substrate and the sequence of engagement of the binding-site residues; previous MD simulations have suggested, however, that positioning of the peptide N-terminus precedes the C-terminus moving into place (*Parker et al., 2021*). Substrate binding then has two distinct effects: first, it exerts a small thermodynamic bias towards the OCC state via increased flexibility in degrees of freedom orthogonal to the overall OF→OCC transition. Second, through engaging R57, substrate entry increases the pKa value of E56 in the ExxER motif, thus thermodynamically facilitating the movement of protons further down the transporter cleft. We can also speculate that-in addition to this thermodynamic favouring of E56 protonation-there might be a kinetic effect on proton transfer from moving the positively charged R57 out of the way of the incoming proton. This could be investigated using reactive MD or QM/MM simulations (both approaches have been employed for other protonation steps of procaryotic peptide transporters, see *Parker et al., 2017* and *Li et al., 2022*), however the putative path is very long ($\approx$1.7 nm between H87 and E56) and may or may not involve a large number of intermediate protonatable residues, in addition to binding site water. While such an investigation is possible in principle, it is beyond the scope of the present study. Likewise, a coupled enhanced-sampling treatment involving both proton movement and large-scale conformational changes-as discussed above for the ordering of steps in the OCC → OF transition-would make for interesting future work, once it becomes computationally tractable.

Our data is not fully determinate with respect to whether protons move to E56 before or after the OF→OCC conformational transition (our CpHMD, for example, remains agnostic on this matter since the shift in the E56 pKa value induced by the substrate is evident in both the OCC and OF states). We may interpret the fact that OCC is raised in energy while H87 is protonated and substrate-induced OCC stabilisation is not found when H87 and D342 are protonated (but does occur when E56 is protonated) as an indication that proton movement is favoured before the transition into OCC is complete. On the other hand, the transition-region interaction between protonated H87 and D317 could also be interpreted as a potential facilitator of the OF→OCC transition. We thus speculate that the proton movement processes may happen as an ensemble of different mechanisms, and potentially occur contemporaneously with the conformational change. This, in addition to a flexible binding pocket, may also contribute to the substrate promiscuity mechanism.

We note at this stage that-throughout our study-we have not investigated the possibility of the substrate C-terminus itself becoming (transiently) protonated. This would need to be taken into account when treating proton movement through the transporter explicitly in the future (see the discussion of such approaches above). There is evidence in our simulations that an additional protonation site-aside from H87, D342, E53, E56, and E622-may be involved in the mechanism, since E622 protonation, while biasing the transporter towards IF, also increases the OCC→IF transition barrier if Ala-Phe substrate is bound (we therefore indicate the proton movements at these stages with a question mark in *Figure 8*). There is thus the intriguing possibility that the substrate itself may temporarily hold the proton, although given the nature of the data presented here only speculation is currently possible on this point. What is clear from our 2D-PMFs, regardless, is that protonation of ExxER glutamates does act as an intracellular gate trigger (and may also pull the transporter through the chemical equilibria all the way from OF). Taking together our 2D-PMFs and ABFE simulations, it is also clear that E622, in addition to being essential for peptide recognition, plays two further roles: its protonation both facilitates substrate release and makes an additive contribution to the IF-directed bias exerted by the intracellular gate triggers (whether E622 forms 'part' of the intracellular gate remains then

as a merely linguistic question). At this stage, we do not yet have an understanding of how exactly intracellular gate opening (which involves breaking an assembly of several hydrophobic residues) is effected by the protonation of these glutamate residues. This question should prove interesting to study in future work. Once the substrate (driven by E622 protonation) and the protons (driven by their electrochemical potential gradient) have left through the open intracellular gate into the intracellular bulk, the resulting apo, standard-protonation-state IF conformation has a thermodynamic preference to return to OCC as evidenced in our 2D-PMFs. We thus arrive back at the starting state and have completed the proton-and substrate-coupled alternating access cycle.

In support of our MD data, we present mutational analysis in cell-based transport assays. Mutations of H87, S321, and D342 to alanine all significantly decrease transport activity, with H87A having the strongest effect. Taken together with similar results in the literature on E53, E56, R57, D317, and E622 (as referenced above), all residues implicated by our study have, therefore, been confirmed their importance for transport via mutagenesis. While the cell-based assay used here cannot differentiate for example between proton-coupling and non-proton-coupling residues, our results still provide a useful first step towards the validation of the gating mechanisms we propose with our PMFs, and should prove informative for the future design of more in-depth experiments.

In conclusion, this study utilises the recent wealth of bacterial and mammalian peptide transporter structures to construct a model of their alternating access mechanism. We explain how the movement of two protons through the transporter drives the accompanying conformational changes, as well as how conformational changes and proton-movement events are coupled to the presence of substrate. Questions regarding some of the finer details, notably the precise sequence of proton movements and conformational transitions (if a single such sequence exists) and whether a further protonation site contributes to the mechanism remain open for future investigation. Nonetheless, the evidence supplied here addresses the alternating access proton-symport mechanism in unprecedented detail, particularly through the extensive use of free-energy simulation techniques. This information will prove useful for the project of employing peptide transporters as vehicles for drug delivery-especially since what determines the efficacy of a transporter substrate is not only related to affinity but crucially also to an ability of the substrate to move through the steps of the alternating access cycle once bound to the transporter.

## Materials and methods
### Definition of tip and base bundle CVs
For the analysis and interpretation of our unbiased MD runs, as well as for the use as a CV for initial metadynamics and umbrella sampling trials, we constructed the tip-CV and base-CV as centre-of-mass distances between the C$\alpha$ atoms of the top and bottom 11 residues of the N-terminal and C-terminus bundles respectively, as illustrated in *Figure 2a*. These residues were picked as the consensus of the DSSP analysis (*Kabsch and Sander, 1983*) of the PepT2 conformational states derived below, listed in *Table 2*.

### MD setup and equilibration
We obtained protein coordinates from cryo-EM for the OF (7NQK) (*Parker et al., 2021*) and IF-partially-occluded (7PMY) (*Killer et al., 2021*) PepT2 conformations, as well as from alphafold 2 (*Jumper et al., 2021*) for the fully-open IF state. We used MODELLER (*Sali and Blundell, 1993*) to fit the rat PepT2 sequence (as used by *Parker et al., 2021*) to the human PepT2 7PMY and alphafold models, using residues 43–409 (TM 1–9) linked as a continuous chain to residues 604–700 (TM 10–12),

**Table 2.** Residue numbers used in the definition of the tip-collective variable (CV) and base-CV.

| Bundle | Residue numbers |
| --- | --- |
| N-terminal bundle tips | 63–74, 79–90, 120–131, 143–154, 194–205, 217–228 |
| N-terminal bundle bases | 46–57, 93–104, 110–121, 161–172, 177–188, 227–238 |
| C-terminal bundle tips | 320–331, 341–352, 392–403, 609–620, 655–666, 671–682 |
| C-terminal bundle bases | 290–301, 359–370, 376–387, 626–637, 642–653, 686–697 |

thereby truncating the extracellular domain as done by *Parker et al., 2021* in their MD simulations. We scored 200 models with QMEANDisCo (*Studer et al., 2020*) and selected the highest-scoring protein model for embedding into a 3:1 POPE:POPG bilayer of target size 10 * 10 nm (210/72 lipid molecules for IF and OCC, 218/72 for OF) with the CHARMM-GUI membrane builder *Wu et al., 2014*. We added ACE/NME capping residues using pymol and solvated the membrane system using GROMACS (*Abraham et al., 2015*) with approximately 21,000 solvent molecules (precise number varies between replicates) at a NaCl concentration of 0.15 M in an orthorhombic box of around 9.9 * 9.9 * 10.8 nm side lengths. Topologies were generated using the AMBER ff14.SB (*Maier et al., 2015*) and slipids (*Jämbeck and Lyubartsev, 2012*) forcefields.

Using the GROMACS MD engine (*Abraham et al., 2015*) in versions 2021.3/2021.4 (the slight version discrepancy is because of different installations on two compute clusters we used), we energy-minimized the systems, assigned initial velocities, and equilibrated with Cα-atom restraints for 200 ps in the NVT ensemble with a leap-frog integrator (using the modified v-rescale thermostat with a stochastic term (*Bussi et al., 2007*) at 310 K throughout our work), then 1 ns in the NPT ensemble (with the berendsen barostat), followed by 20 ns of further Cα-restrained NPT equilibration (using the Parrinello-Rahman barostat *Parrinello and Rahman, 1981*, as for all subsequent production runs).

We obtained Ala-Phe dipeptide-bound boxes by aligning the holo PepT1 cryo-EM structure (7PMW) onto our equilibrated PepT2 MD boxes, copying the ligand coordinates, and repeating the same equilibration protocol as before (where the peptide substrate Cα atoms were also restrained). The peptide ligand was parametrised using AMBER ff14.SB (*Maier et al., 2015*).

## Derivation of OF, OCC, and IF conformational states

As shown in *Figure 1—figure supplement 2a*, the IF-partially-occluded structure (7PMY) does not behave well in MD (1 μs production runs from triplicate embeddings), since it either partially opens its extracellular gate (replicates 2–3) or partial helical unfolding in the intracellular gate occurs due to hydrophobic collapse (rep 1). This may be due to a variety of factors; one possibility is instability in the protein following the removal of the bound substrate in our simulations. In contrast (*Figure 1—figure supplement 2b*), embedding replicates 1 and 3 of the alphafold IF state behave well. We picked the end-coordinates of replicate 3 as our IF state, due to the wider opening of its intracellular gate. We then sought to derive an OCC state via MD from replicate 1, see the paragraph below. We also note that embeddings from the OF cryo-EM structure (7NQK) remain stable in the OF conformation, we picked replicate 1 for our work.

To derive an OCC state from an IF box, we conducted five replicates of well-tempered metadynamics (*Barducci et al., 2008*) as implemented in PLUMED 2.7 (*Tribello et al., 2014*) along the base-bundle CV (see *Figure 2a*), using eight walkers, hill height 1 kJ mol$^{-1}$ with sigma 0.022 nm deposited every 500 steps, and a bias factor of 100. From the resulting set of trajectories, we picked frames around the mark of 20 ns simulation time and a base-CV value of around 2.0 nm (we chose these values based on visual inspection of the trajectories, where we noticed that base-CV values significantly below 2.0 nm lead to artefacts such as partial unfolding of the ends of helices, as did continuing the metadynamics simulations for longer than necessary to obtain the desired states). We ran triplicate 100 ns-long unbiased MD from the obtained states for each of the five replicates, and found the OCC state obtained from the first replicate to be stably situated within the range of base-CV values observed in the OF-state trajectories. The micro-second-long unbiased MD runs as well as our 2D-PMFs confirm that this protein conformation is a stable basin, and that different protonation states of key residues can drive its opening to either the IF or OF states. While this does not rule out the existence of different OCC states, it confirms the properties of the conformation we found as a functional OCC state.

## Unbiased MD of the OCC state

Unbiased MD runs of the OCC state in different conditions were conducted in triplicates using the same simulation parameters as for the long Cα-restrained equilibrations described in the section on equilibration, but removing all restraints. The starting coordinates were—for the first replicate—the OCC state derived as described in the foregoing section, and the second and third replicates were initialised from the 500 ns and the final frame of the first replicate trajectory. Protonation states changes and mutations were carried out using PyMOL and GROMACS pdb2gmx independently for each replicate,

followed by re-running the full equilibration protocol for all new boxes. Taken together, we conducted unbiased MD for 24 conditions, giving 72 μs of production sampling.

## Metadynamics and steered MD

For our initial trials of enhanced sampling on PepT2 conformational changes-which showed hysteretic behaviour (see *Figure 2—figure supplement 3*), we attempted steered MD (SMD) and metadynamics for the OCC↔OF transition in the WT, unprotonated state.

Two instances of eight-walker well-tempered metadynamics were run, starting from the OF and OCC states, biased along the tip-CV with hills of height 1 kJ mol$^{-1}$ and sigma 0.0455 nm deposited every 500 steps, using a bias factor of 100. A harmonic flat-bottom restraint with boundaries of 2.0–3.0 nm and force constant $5 \times 10^4$ kJ mol$^{-1}$ nm$^{-2}$ was applied on the CV value. Sampling was run for 108 ns per walker for the simulations starting from OF, and 213 ns per walker starting from OCC.

To generate paths for umbrella sampling, SMD was run starting at OF towards OCC and vice versa with the heavy-atom RMSD to the target conformation as CV, using a harmonic potential centered to zero RMSD with a force constant sliding from 0 up to $2.5 \times 10^5$ kJ mol$^{-1}$ nm$^{-2}$ over 200 ns. The harmonic potential was then switched off over 2 ns, followed by 48 ns of unbiased MD. We then projected the SMD trajectories onto the tip-CV, picked 48 frames spaced equally along the CV and performed 1D replica-exchange umbrella sampling (REUS) using a force constant of $3 \times 10^4$ kJ mol$^{-1}$ nm$^{-2}$, for 92 ns per window for the OCC→OF derived boxes and 127 ns per window in the reverse direction. A total of 13.6 μs of MD time was thus expended on these trials.

## MEMENTO path generation

We have recently proposed the MEMENTO method for history-independent path generation between given end-states *Lichtinger and Biggin, 2023*.

In short, protein coordinates are morphed, followed by reconstructing an ensemble of structures at each morphing intermediate using MODELLER. Monte-Carlo simulated annealing with an energy function based on between-intermediate RMSD values then finds a smooth path through these ensembles. For membrane proteins, lipids are taken from the end-state that occupies a larger area in the membrane (in this case, OF for the OCC↔OF transition, and IF for OCC↔IF), and fitted around the new protein coordinates by expanding the membrane, followed by iterative compression and energy minimization. Ligands are not morphed but translated, interpolating between the ligand centres of masses in the end-states, and then equilibrated in the protein structure using MD. MEMENTO is implemented as the PyMEMENTO package (https://github.com/simonlichtinger/PyMEMENTO, copy archived at *Lichtinger, 2024*), we provide example scripts for its usage on PepT2 in the supplementary data.

In this study, we ran MEMENTO with 24 windows in triplicates for both the OCC↔OF and OCC↔IF transitions in different protein protonation and mutation states, and in the presence or absence of ligands. The apo and holo MEMENTO replicates were initialised from the 0 ns, 500 ns, and 1000 ns frames of the first replicate of the 1 μs unbiased MD run for each conformational state (using always the unprotonated, WT condition, but apo/holo trajectories, respectively). Protonation state changes and mutations were then carried out using the built-in functionality of PyMEMENTO, and equilibrated at each intermediate state for 90 ns. The total MD simulation time spent on equilibrations as part of the MEMENTO method across the 22 sampled conditions was 47 μs.

**Table 3.** Overview of all 1D-PMF sampling.

| Condition | Simulation time/ns |
| --- | --- |
| OCC↔OF, standard prot | 24 * (266+244+244) |
| OCC↔OF, H87 & D342 prot | 24 * (327+244+244) |
| OCC↔IF, standard prot | 24 * (242+248+246) |
| OCC↔IF, E53 prot | 24 * (158+155+154) |
| Total | 67 μs |

## 1D-PMF calculations

Starting with the triplicate equilibrated MEMENTO boxes for the (all apo) OCC↔OF standard protonation and H87 & D342 protonated states, as well as OCC↔IF standard protonation and E53 protonated states, we ran 1D-replica exchange umbrella sampling (REUS, exchange every 1000 steps; using PLUMED 2, *Tribello et al., 2014*) along the tip-and base-CVs, respectively, using a force constant of $4 \times 10^3$ kJ mol⁻¹ nm⁻². The amount of sampling collected in each case is summarised in *Table 3*.

## 2D-CV derivation

Using the trajectory data from our 1D-PMFs, we derived 2D CVs via a PCA-based approach we have previously described for LEUT (*Lichtinger and Biggin, 2023*). We pooled all sampling collected in 1D-REUS runs along the tip-CV for apo OCC↔OF (and equivalently for OCC↔IF. The same procedure was taken for these trajectories, and we will only explicitly write about OCC↔OF in the following paragraph). Using GROMACS tools, we ran PCA of the Cα positions of residues contained in the transmembrane region (see *Table 2* above). The first principal component (PC) accounts for 50% of the variance; adding an extra 15 PCs increases coverage to 78% of the variance (comparable to our results on LEUT). The first PC (see *Figure 2—figure supplement 5*) describes the gating motions of the respective conformational changes, behaving similarly to the tip CV (or base CV)-expectedly so, given it was the CV used in our 1D-REUS. To explain differences between replicates (see *Figure 2—figure supplement 4a and c*), we used differential evolution (*Storn and Price, 1997*) as implemented in scipy (*Virtanen et al., 2020*) to maximise an entropy-like metric of distances between MEMENTO path frames for linear combinations of the PCs 2–16:

$$\frac{2}{N_{\text{rep}} * (N_{\text{rep}} - 1)} \sum_{i=1}^{N_{\text{rep}}} \sum_{j=i+1}^{N_{\text{rep}}} \sum_{n}^{N_{\text{windows}}} log(\sqrt{\sum_{\text{PC}} (\mathbf{X}(n,i) - \mathbf{X}(n,j))^2}), \tag{1}$$

where $N_{\text{rep}}$ is the number of replicates and by $\mathbf{X}(n,i) - \mathbf{X}(n,j)$ we denote the distance between two conformational frames in different replicates $i$ and $j$, evaluated in a projection along a given combination of principal components. The result was termed PC 2 henceforth for simplicity and used as the second CV in 2D-REUS (*Figure 2—figure supplement 5*).

## 2D-PMF calculations

Using the same equilibrated MEMENTO paths as above and the 2D-CVs we derived, we calculated 2D-PMFs of the OCC↔OF and OCC↔IF transitions in several protonation/mutation states, with and without Ala-Phe substrate-bound, using 2D-REUS. As shown in *Figure 3—figure supplement 5*, we found that a lower force constant of $2 \times 10^6$ kJ mol⁻¹ nm⁻² leads to good histogram overlap in the lower-lying regions of the PMF, but has poor overlap near the transition region. In turn, a higher force constant of $1 \times 10^7$ kJ mol⁻¹ nm⁻² gives good window overlap in the transition region while not sampling broadly enough in large basins. Therefore, for each condition and MEMENTO replicate, we ran windows at both force constants and included them all in the WHAM analysis, thus ensuring sufficient sampling through the CV space. Replica-exchange was run within the 24 windows corresponding to each MEMENTO replicate–force constant combination, and a total of 144 windows contribute to each 2D-PMF.

The sampling collected in all conditions is detailed in *Table 4*, aiming for between 180 and 210 ns per window though exact amounts differ with heterogeneous hardware and slightly different box sizes.

## Absolute binding free energies

To probe the affinity of the Ala-Phe substrate to the PepT2 OF and IF conformations in different protonation states, we conducted ABFE simulations (*Aldeghi et al., 2016*; *Aldeghi et al., 2018*) in gromacs, using an equilibrium approach. For this, we changed our mdp files to use the stochastic dynamics integrator (doubling as a thermostat) and set the relevant free-energy flags, including soft-core van-der-Waals interactions (alpha = 0.5, power = 1, sigma = 0.3) and the couple-intramol=yes flag for consistency with larger ligands in other work. Our lambda-protocol was to first add Boresch restraints (*Boresch et al., 2003*) (for the complex thermodynamic leg only, ligand side was calculated using the analytic formula; through values 0, 0.01, 0.025, 0.05, 0.075, 0.1, 0.2, 0.3, 0.4, 0.5, 0.6, 0.8,

**Table 4.** Overview of all 2D-PMF sampling.

| Condition | Simulation time/ns |
|---|---|
| OCC↔OF, standard prot | 24 * (233+195+195+208+208+207) |
| OCC↔OF, H87 prot | 24 * (230+200+188+201+228+229) |
| OCC↔OF, D342 prot | 24 * (204+232+204+224+229+207) |
| OCC↔OF, H87&D342 prot | 24 * (196+196+195+194+194+201) |
| OCC↔OF, D342A | 24 * (182+183+206+181+183+183) |
| OCC↔OF, H87A | 24 * (176+182+183+184+182+177) |
| OCC↔OF, E53 prot | 24 * (185+180+209+179+218+213) |
| OCC↔OF, E56 prot | 24 * (194+194+194+192+193+193) |
| OCC↔OF, R206D&D342R | 24 * (227+213+183+222+211+180) |
| OCC↔OF, holo, standard prot | 24 * (200+196+195+200+194+201) |
| OCC↔OF, holo, E53 prot | 24 * (185+180+210+179+218+213) |
| OCC↔OF, holo, E56 prot | 24 * (194+194+194+192+193+193) |
| OCC↔OF, holo, H87&D342 prot | 24 * (198+191+198+190+182+201) |
| OCC↔IF, standard prot | 24 * (197+216+203+192+197+199) |
| OCC↔IF, E53 prot | 24 * (197+208+199+192+193+195) |
| OCC↔IF, E622 prot | 24 * (199+201+200+196+217+197) |
| OCC↔IF, E53&E622 prot | 24 * (193+237+240+187+250+200) |
| OCC↔IF, holo, standard prot | 24 * (190+195+191+196+192+197) |
| OCC↔IF, holo, E53 prot | 24 * (189+196+197+199+189+198) |
| OCC↔IF, holo, E622 prot | 24 * (194+191 + 206+197 + 198+199) |
| OCC↔IF, holo, E53&E622 prot | 24 * (197+198 + 180+195 + 190+204) |
| Total | 598 µs |

1.0), then annihilate coulomb interactions (even 0.1 spacings) followed by vdw interactions (even 0.05 spacings). We equilibrated for 200 ps of NVT and 1 ns of NPT at each lambda window, and ran production simulations with replica-exchange attempts every 1000 steps for 30 ns per window on the complex thermodynamic leg and 100 ns on the ligand-only leg.

The Boresch restraints for ABFE simulations were obtained MDRestraintsGenerator (*Alibay et al., 2022*) by running the restraint finding algorithm over the ≈200 ns 2D-REUS trajectory at the relevant conformation and protonation state, and we used the trajectory frame closest to the restraint centre as input for subsequent ABFE. This was done for each of the triplicate MEMENTO runs, giving three candidate ligand binding poses. For the pose that was found to have the highest single-replicate affinity, four replicates of unbiased, restraint-free 200 ns-long equilibrations were also started from the frames, processed with MDRestraintsGenerator and used to make replicates for ABFE runs to give an error estimate as mean plus-minus standard deviation. By following this protocol for the OF, OF E56 prot, IF, and IF E622 prot conditions, we sampled for a total of 4 conditions * 7 boxes per condition * 44 windows * 31.2 ns = 38 µs.

## Constant-pH MD

To probe the substrate-dependence of the E53 and E56 pKa values, we ran constant pH simulations (CpHMD) using the hybrid solvent approach with discrete protonation states (*Swails et al., 2014*) as implemented in the AMBER software (*Case and Aktulga, 2022*). We took the MEMENTO starting frames from above as triplicate initial coordinates of OF and OCC states in the presence and absence

of Ala-Phe. We then used tleap and in-house scripts to convert our boxes to the AMBER constant pH forcefield fork for protein, substrate, and solutes and to the lipid21 force field (*Dickson et al., 2022*) for the membrane. We prepared constant pH simulations as in the tutorial by , and ran them for 1 µs at 8 pH replica windows (pH 0–7), in the NVT ensemble with a langevin thermostat (at 310 K as before), attempting protonation state changes every 100 steps, running 100 steps of relaxation dynamics for every exchange, and attempting replica exchange every 1000 steps. Analysis was performed using the cphstats programme and in-house scripts for fitting titration curves. In *Figure 6*, analysis is performed per replicate, reporting mean ± standard deviation for each condition and residue. In *Figure 6—figure supplement 1 and 2*, we show pKa values estimated over simulation time from 10 ns chunks of all CpHMD runs. We also analyse this data in terms of histograms of the chunk-estimated pKa values, pooling all data for each condition and residue.

A total of 4 conditions * 3 replicates * 8 windows * 1 µs=96 µs of sampling were thus collected.

## Cell-based transport assays

Transport assays were carried out using a modified version of the protocol by *Parker et al., 2021*. Human cervical adenocarcinoma HeLa cells were purchased from Merck 93021013 and confirmed by STR profiling. The cell line was mycoplasma negative, as confirmed using the EZ-PCR Mycoplasma Test Kit (K1-0210, Geneflow). H. Hela cells were cultured in DMEM + GlutaMAX medium, supplemented with 10% FBS. 12-well plates were prepared by seeding $9 \times 10^4$ cells per well in 1 mL of medium, and transfected after 24 hr with 0.8 µg of PepT2-constructs in pEF5-FRT-eGFP vector (or 0.5 µg of insert vector + 0.3 µg of empty vector, where specified), with 1.6 µg of fugene transfection reagent. The medium was exchanged 24 hr post-transfection, and assays were carried out 40 hr post transfection. Cells were washed two times with ≈0.6 mL of assay buffer (20 mM HEPES pH 7.5, 120 mM NaCl, 2 mM $MgSO_4$, and 25 mM glucose), then incubated with 0.3 mL assay buffer containing 20 mM $\beta$-ala-lys-AMCA substrate for 15 min. The cells were then washed three times with ≈0.6 mL of assay buffer, and incubated with 0.25 mL of lysis buffer (20 mM Trist pH 7.5+0.2% Triton x-100) for 5 min. The fluorescence (340 nm excitation, 460 nm read-out) of 0.15 mL of the lysate was normalised by the protein amount in each well (as determined from BCA assay of 20 µL lysate). We removed two outliers from the WT (0.8 µg) transport assay dataset, giving n=46. The data was then scaled to the mean WT (0.8 µg) transport level as 100%.

## Protein expression controls

For comparing PepT2 WT and mutant expression levels, Hela cells were seeded in six-well plates at $1.8 \times 10^5$ cells per well in 2 mL of medium. Transfection was after 24 hr with 1.6 µg of PepT2-constructs in pEF5-FRT-eGFP vector (or 1.0 µg of insert vector +0.6 µg of empty vector, where specified) and 3.2 µg of fugene; the medium was exchanged 24 hr post-transfection. The cells were washed three times with ≈0.6 mL of PBS, harvested using 0.1% trypsin, pelleted, re-suspended in 100 µL PBS with protease inhibitor and lysed through 3 x freeze-thawing. The lysates from three wells were pooled for each mutant to increase between-sample consistency. 4.5 µL of each sample were loaded onto a 10% SDS-PAGE gel, and western blot was performed using an anti-GFP antibody. The membrane was then stripped and developed again with an anti-$\beta$-actin antibody to control for gel loading.

## Protein localisation controls

To confirm the plasma membrane localisation of PepT2 WT and mutants, we seeded $1.8 \times 10^5$ Hela cells per well in 2 mL of medium in a six-well plate with added coverslips. Transfection was as for the expression controls 24 hr after seeding. 20 hr post-transfection, the cells were washed three times with ≈0.6 mL of PBS, fixed with PFA for 10 min at room temperature, washed three times, incubated with PBS + 50 mM $NH_4Cl$ for 10 min, washed three times, incubated with PBS + 0.1% Triton x-100 for 5 min, washed three times, stained with DAPI, washed five times, and mounted on slides with Immu-Mount. Images were recorded in the GFP and DAPI channels.

## Acknowledgements

We would like to thank Dr. Zhiyi Wu for general training in molecular dynamics methodology and help in the early stages of this project, Dr. Irfan Alibay for training in ABFE simulations, Dr. Gabriel Kuteyi for training in mammalian cell culture techniques and Sacha Salphati for training in fluorescent microscopy,

as well as all members of the Biggin and Newstead groups for helpful discussions. This project was funded by the Wellcome Trust (Grant ID: 218514/Z/19/Z). Compute resources were also provided by the EPSRC ARCHER2, Jade 2 and N8 CIR BEDE facilities, granted via the High-End Computing Consortium for Biomolecular Simulation (HECBioSim), supported by EPSRC (EP/X035603/1).

## Additional information

### Funding

| Funder | Grant reference number | Author |
|---|---|---|
| Wellcome Trust | 218514/Z/19/Z | Simon M Lichtinger<br>Simon Newstead<br>Philip Biggin |
| Engineering and Physical Sciences Research Council | EP/X035603/1 | Philip C Biggin |

The funders had no role in study design, data collection and interpretation, or the decision to submit the work for publication. For the purpose of Open Access, the authors have applied a CC BY public copyright license to any Author Accepted Manuscript version arising from this submission.

### Author contributions
Simon M Lichtinger, Formal analysis, Investigation, Methodology, Writing - original draft; Joanne L Parker, Formal analysis, Investigation, Methodology, Writing – review and editing; Simon Newstead, Conceptualization, Supervision, Funding acquisition, Project administration, Writing – review and editing; Philip C Biggin, Conceptualization, Resources, Supervision, Funding acquisition, Project administration, Writing – review and editing

### Author ORCIDs
Simon Newstead ⬥ https://orcid.org/0000-0001-7432-2270
Philip C Biggin ⬥ http://orcid.org/0000-0001-5100-8836

Reviewer #1 (Public review): https://doi.org/10.7554/eLife.96507.3.sa1
Reviewer #2 (Public review): https://doi.org/10.7554/eLife.96507.3.sa2
Reviewer #3 (Public review): https://doi.org/10.7554/eLife.96507.3.sa3
Author response https://doi.org/10.7554/eLife.96507.3.sa4

## Additional files

### Supplementary files
• MDAR checklist

### Data availability
Simulation and experimental data produced in this work is available at https://doi.org/10.5281/zenodo.10561418. This includes key coordinate files and python scripts, as well as simulation trajectories projected onto the CVs of interest in plumed output format, and relevant processed files for PMF, pKa and ABFE calculations. This should sufficient to reproduce the work here. Sharing full trajectory data by default is currently not practical since the simulation data accumulated as production runs during this study totals around 7 TB (for comparison, zenodo imposes a maximum limit of 50 GB). We do already share trajectory projections onto CVs used in free energy calculations (amounting to around 14GB), which will enable other researchers to reanalyse the simulation runs as far as convergence is concerned. Should a structural reanalysis be desired, we will be happy to share the subset of trajectory data required for the analysis in question and coordinate with the requesting researcher regarding how to best achieve the transfer logistically. The PyMEMENTO software is freely available at https://github.com/simonlichtinger/PyMEMENTO (copy archived at *Lichtinger, 2024*).

The following dataset was generated:

| Author(s) | Year | Dataset title | Dataset URL | Database and Identifier |
|-----------|------|---------------|-------------|-------------------------|
| Lichtinger S, Parker J, Newstead S, Biggin PC | 2024 | Supplementary data for: The mechanism of mammalian proton-coupled peptide transporters | https://doi.org/10.5281/zenodo.10561418 | Zenodo, 10.5281/zenodo.10561418 |

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
