## [Editor Report · eLife assessment]

This study provides **important** insight into the mechanisms of proton-coupled oligopeptide transporters. It uses enhanced-sampling molecular dynamics (MD), backed by cell-based assays, revealing the importance of protonation of selected residues for PepT2 function. The simulation approaches are **convincing**, using long MD simulations, constant-pH MD and free energy calculations. Overall, the work has led to findings that will appeal to structural biologists, biochemists, and biophysicists studying membrane transporters.

---

## [Referee Report · Reviewer #1 (Public review)]

The authors have performed all-atom MD simulations to study the working mechanism of hsPepT2. It is widely accepted that conformational transitions of proton-coupled oligopeptide transporters (POTs) are linked with gating hydrogen bonds and salt bridges involving protonatable residues, whose protonation triggers gate openings. Through unbiased MD simulations, the authors identified extra-cellular (H87 and D342) and intra-cellular (E53 and E622) triggers. The authors then validated these triggers using free energy calculations (FECs) and assessed the engagement of the substrate (Ala-Phe dipeptide). The linkage of substrate release with the protonation of the ExxER motif (E53 and E56) was confirmed using constant-pH molecular dynamics (CpHMD) simulations and cell-based transport assays. An alternating-access mechanism was proposed. The study was largely conducted properly, and the paper was well-organized. However, I have a couple of concerns for the authors to consider addressing.

(1) As a proton-coupled membrane protein, the conformational dynamics of hsPepT2 is closely coupled to protonation events of gating residues. Instead of using semi-reactive methods like CpHMD or reactive methods such as reactive MD, where the coupling is accounted for, the authors opted for extensive non-reactive regular MD simulations to explore this coupling. Note that I am not criticizing the choice of methods, and I think those regular MD simulations were well designed and conducted. But I do have two concerns.

(a) Ideally, proton-coupled conformational transitions should be modelled using a free energy landscape with two or more reaction coordinates (or CVs), with one describing the protonation event and the others describing the conformational transitions. The minimum free energy path then illustrates the reaction progress, such as OCC/H87D342- ↔ OCC/H87HD342H ↔ OF/H87HD342H as displayed in Figure 3. Without including the protonation as a CV, the authors tried to model the free energy changes from multiple FECs using different charge states of H87 and D342. This is a practical workaround, and the conclusion drawn (the OCC↔OF transition is downhill with protonated H87 and D342) seems valid. However, I don't think the OF states with different charge states (OF/H87D342-, OF/H87HD342-, OF/H87D342H, and OF/H87HD342H) are equally stable, as plotted in Figure 3b. The concern extends to other cases like Figures 4b, S7, S10, S12, S15, and S16. While it may be appropriate to match all four OF states in the free energy plot for comparison purposes, the authors should clarify this to ensure readers are not misled.

(b) Regarding the substrate impact, it appears that the authors assumed fixed protonation states. I am afraid this is not necessarily the case. Variations in PepT2 stoichiometry suggests that substrates likely participate in proton transport, like the Phe-Ala (2:1) and Phe-Gln (1:1) dipeptides mentioned in the introduction. And it is not rigorous to assume that the N- and C-termini of a peptide do not protonate/deprotonate when transported. I think the authors should explicitly state that the current work and the proposed mechanism (Figure 8) are based on the assumption that the substrates do not uptake/release proton(s).

(2) I have more serious concerns about the CpHMD employed in the study.

(a) The CpHMD in AMBER is not rigorous for membrane simulations. The underlying generalized Born model fails to consider the membrane environment when updating charge states. In other words, the CpHMD places a membrane protein in a water environment to judge if changes in charge states are energetically favorable. While this might not be a big issue for peripheral residues of membrane proteins, it is likely unphysical for internal residues like the ExxER motif. As I recall, the developers have never used the method to study membrane proteins themselves. The only CpHMD variant suitable for membrane proteins is the membrane-enabled hybrid-solvent CpHMD in CHARMM. While I do not expect the authors to redo their CpHMD simulations, I do hope the authors recognize the limitation of their method.

(b) It appears that the authors did not make the substrate (Ala-Phe dipeptide) protonatable in holo-simulations. This oversight prevents a complete representation of ligand-induced protonation events, particularly given that the substrate ion-pairs with hsPepT2 through its N- & C-termini. I believe it would be valuable for the authors to acknowledge this potential limitation.

---

## [Referee Report · Reviewer #2 (Public review)]

Summary:

This is an interesting manuscript that describes a series of molecular dynamics studies on the peptide transporter PepT2 (SLC15A2). They examine, in particular, the effect on the transport cycle of protonation of various charged amino acids within the protein. They then validate their conclusions by mutating two of the residues that they predict to be critical for transport in cell-based transport assays. The study suggests a series of protonation steps that are necessary for transport to occur in Petp2. Comparison with bacterial proteins from the same family show that while the overall architecture of the proteins and likely mechanism are similar, the residues involved in the mechanism may differ.

Strengths:

This is an interesting and rigorous study that uses various state of the art molecular dynamics techniques to dissect the transport cycle of PepT2 with nearly 1ms of sampling. It gives insight into the transport mechanism, investigating how protonation of selected residues can alter the energetic barriers between various states of the transport cycle. The authors have, in general, been very careful in their interpretation of the data.

Weaknesses:

Interestingly, they suggest that there is an additional protonation event that may take place as the protein goes from occluded to inward-facing (clear from Figure 8) but as the authors comment they have not identified this residue(s).

---

## [Referee Report · Reviewer #3 (Public review)]

Summary:

Lichtinger et al. have used an extensive set of molecular dynamics (MD) simulations to study the conformational dynamics and transport cycle of an important member of the proton-coupled oligopeptide transporters (POTs), namely SLC15A2 or PepT2. This protein is one of the most well-studied mammalian POT transporters that provides a good model with enough insight and structural information to be studied computationally using advanced enhanced sampling methods employed in this work. The authors have used microsecond-level MD simulations, constant-PH MD, and alchemical binding free energy calculations along with cell-based transport assay measurements; however, the most important part of this work is the use of enhanced sampling techniques to study the conformational dynamics of PepT2 under different conditions.

The study attempts to identify links between conformational dynamics and chemical events such as proton binding, ligand-protein interactions, and intramolecular interactions. The ultimate goal is of course to understand the proton-coupled peptide and drug transport by PepT2 and homologous transporters in the solute carrier family.

Some of the key results include (1) Protonation of H87 and D342 initiate the occluded (Occ) to the outward-facing (OF) state transition; (2) In the OF state, through engaging R57, substrate entry increases the pKa value of E56 and thermodynamically facilitates the movement of protons further down; (3) E622 is not only essential for peptide recognition but also its protonation facilitates substrate release and contributes to the intracellular gate opening. In addition, cell-based transport assays show that mutation of residues such as H87 and

D342 significantly decrease transport activity as expected from simulations.

Strengths:

(1) This is an extensive MD based study of PepT2, which is beyond the typical MD studies both in terms of the sheer volume of simulations as well as the advanced methodology used. The authors have not limited themselves to one approach and have appropriately combined equilibrium MD with alchemical free energy calculations, constant-pH MD and geometry-based free energy calculations. Each of these 4 methods provides a unique insight regarding the transport mechanism of PepT2.

(2) The authors have not limited themselves to computational work and has performed experiments as well. The cell-based transport assays clearly establish the importance of the residues that have been identified as significant contributors to the transport mechanism using simulations.

(3) The conclusions made based on the simulations are mostly convincing and provide useful information regarding the proton pathway and the role of important residues in proton binding, protein-ligand interaction, and conformational changes.

Weaknesses:

There are inherent limitations with the methodology used such as the MEMENTO and constant pH MD that have been briefly noted in the manuscript.

---

## [Author Response]

The following is the authors’ response to the original reviews.

**eLife assessment**
This study provides valuable information on the mechanism of PepT2 through enhanced-sampling molecular dynamics, backed by cell-based assays, highlighting the importance of protonation of selected residues for the function of a proton-coupled oligopeptide transporter (hsPepT2). The molecular dynamics approaches are convincing, but with limitations that could be addressed in the manuscript, including lack of incorporation of a protonation coordinate in the free energy landscape, possibility of protonation of the substrate, errors with the chosen constant pH MD method for membrane proteins, dismissal of hysteresis emerging from the MEMENTO method, and the likelihood of other residues being affected by peptide binding. Some changes to the presentation could be considered, including a better description of pKa calculations and the inclusion of error bars in all PMFs. Overall, the findings will appeal to structural biologists, biochemists, and biophysicists studying membrane transporters.

We would like to express our gratitude to the reviewers for providing their feedback on our manuscript, and also for recognising the variety of computational methods employed, the amount of sampling collected and the experimental validation undertaken. Following the individual reviewer comments, as addressed point-by-point below, we have prepared a revised manuscript, but before that we address some of the comments made above in the general assessment:

“lack of incorporation of a protonation coordinate in the free energy landscape”.

We acknowledge that of course it would be highly desirable to treat protonation state changes explicitly and fully coupled to conformational changes. However, at this point in time, evaluating such a free energy landscape is not computationally feasible (especially considering that the non-reactive approach taken here already amounts to almost 1ms of total sampling time). Previous reports in the literature tend to focus on either simpler systems or a reduced subset of a larger problem. As we were trying to obtain information on the whole transport cycle, we decided to focus here on non-reactive methods.

“possibility of protonation of the substrate”.

The reviewers are correct in pointing out this possibility, which we had not discussed explicitly in our manuscript. Briefly, while we describe a mechanism in which protonation of only protein residues (with an unprotonated ligand) can account for driving all the necessary conformational changes of the transport cycle, there is some evidence for a further intermediate protonation site in our data (as we commented on in the first version of the manuscript as well), which may or may not be the substrate itself. A future explicit treatment of the proton movements through the transporter, when it will become computationally tractable to do so, will have to include the substrate as a possible protonation site; for the present moment, we have amended our discussion to alert the reader to the possibility that the substrate could be an intermediate to proton transport. This has repercussions for our study of the E56 pKa value, where – if protons reside with a significant population at the substrate C-terminus – our calculated shift in pKa upon substrate binding could be an overestimate, although we would qualitatively expect the direction of shift to be unaffected. However, we also anticipate that treating this potential coupling explicitly would make convergence of any CpHMD calculation impractical to achieve and thus it may be the case that for now only a semi-quantitative conclusion is all that can be obtained.

“errors with the chosen constant pH MD method for membrane proteins”.

We acknowledge that – as reviewer #1 has reminded us – the AMBER implementation of hybrid-solvent CpHMD is not rigorous for membrane proteins, and as such added a cautionary note to our paper. We also explain how the use of the ABFE thermodynamic cycle calculations helps to validate the CpHMD results in a completely orthogonal manner (we have promoted this validation, which was in the supplementary figures, into the main text in the revised version). We therefore remain reasonably confident in the results presented with regards to the reported pKa shift of E56 upon substrate binding, and suggest that if the impact of neglecting the membrane in the implicit-solvent stage of CpHMD is significant, then there is likely an error cancellation when considering shifts induced by the incoming substrate.

“dismissal of hysteresis emerging from the MEMENTO method”.

We have shown in our method design paper how the use of the MEMENTO method drastically reduces hysteresis compared to steered MD for path generation, and find this improvement again for PepT2 in this study. We address reviewer #3’s concern about our presentation on this point by revising our introduction of the MEMENTO method, as detailed in the response below.

“the likelihood of other residues being affected by peptide binding”.

In this study, we have investigated in detail the involvement of several residues in proton-coupled di-peptide transport by PepT2. Short of the potential intermediate protonation site mentioned above, the set of residues we investigate form a minimal set of sorts within which the important driving forces of alternating access can be rationalised. We have not investigated in substantial detail here the residues involved in holding the peptide in the binding site, as they are well studied in the literature and ligand promiscuity is not the problem of interest here. It remains entirely possible that further processes contribute to the mechanism of driving conformational changes by involving other residues not considered in this paper. We have now made our speculation that an ensemble of different processes may be contributing simultaneously more explicit in our revision, but do not believe any of our conclusions would be affected by this.

As for the additional suggested changes in presentation, we provide the requested details on the CpHMD analysis. Furthermore, we use the convergence data presented separately in figures S12 and S16 to include error bars on our 1D-reprojections of the 2D-PMFs in figures 3, 4 and 5. (Note that we have opted to not do so in figures S10 and S15 which collate all 1D PMF reprojections for the OCC ↔ OF and OCC ↔ IF transitions in single reference plots, respectively, to avoid overcrowding those necessarily busy figures). We have also changed the colours schemes of these plots in our revision to improve accessibility. We have additionally taken the opportunity to fix some typos and further clarified some other statements throughout the manuscript, besides the requests from the reviewers.

**Reviewer #1 (Public Review):**
The authors have performed all-atom MD simulations to study the working mechanism of hsPepT2. It is widely accepted that conformational transitions of proton-coupled oligopeptide transporters (POTs) are linked with gating hydrogen bonds and salt bridges involving protonatable residues, whose protonation triggers gate openings. Through unbiased MD simulations, the authors identified extra-cellular (H87 and D342) and intra-cellular (E53 and E622) triggers. The authors then validated these triggers using free energy calculations (FECs) and assessed the engagement of the substrate (Ala-Phe dipeptide). The linkage of substrate release with the protonation of the ExxER motif (E53 and E56) was confirmed using constant-pH molecular dynamics (CpHMD) simulations and cellbased transport assays. An alternating-access mechanism was proposed. The study was largely conducted properly, and the paper was well-organized. However, I have a couple of concerns for the authors to consider addressing.

We would like to note here that it may be slightly misleading to the reader to state that “The linkage of substrate release with the protonation of the ExxER motif (E53 and E56) was confirmed using constant-pH molecular dynamics (CpHMD) simulations and cell-based transport assays.” The cellbased transport assays confirmed the importance of the extracellular gating trigger residues H87, S321 and D342 (as mentioned in the preceding sentence), not of the substrate-protonation link as this line might be understood to suggest.

(1) As a proton-coupled membrane protein, the conformational dynamics of hsPepT2 are closely coupled to protonation events of gating residues. Instead of using semi-reactive methods like CpHMD or reactive methods such as reactive MD, where the coupling is accounted for, the authors opted for extensive non-reactive regular MD simulations to explore this coupling. Note that I am not criticizing the choice of methods, and I think those regular MD simulations were well-designed and conducted. But I do have two concerns.a) Ideally, proton-coupled conformational transitions should be modelled using a free energy landscape with two or more reaction coordinates (or CVs), with one describing the protonation event and the other describing the conformational transitions. The minimum free energy path then illustrates the reaction progress, such as OCC/H87D342- → OCC/H87HD342H → OF/H87HD342H as displayed in Figure 3.

We concur with the reviewer that the ideal way of describing the processes studied in our paper would be as a higher-dimensional free energy landscapes obtained from a simulation method that can explicitly model proton-transfer processes. Indeed, it would have been particularly interesting and potentially informative with regards to the movement of protons down into the transporter in the OF → OCC → IF sequence of transitions. As we note in our discussion on the H87→E56 proton transfer:

“This could be investigated using reactive MD or QM/MM simulations (both approaches have been employed for other protonation steps of prokaryotic peptide transporters, see Parker et al. (2017) and Li et al. (2022)). However, the putative path is very long (≈ 1.7 nm between H87 and E56) and may or may not involve a large number of intermediate protonatable residues, in addition to binding site water. While such an investigation is possible in principle, it is beyond the scope of the present study.”

Where even sampling the proton transfer step itself in an essentially static protein conformation would be pushing the boundaries of what has been achieved in the field, we believe that considering the current state-of-the-art, a fully coupled investigation of large-scale conformational changes and proton-transfer reaction is not yet feasible in a realistic/practical time frame. We also note this limitation already when we say that:

“The question of whether proton binding happens in OCC or OF warrants further investigation, and indeed the co-existence of several mechanisms may be plausible here”.

Nonetheless, we are actively exploring approaches to treat uptake and movement of protons explicitly for future work.

In our revision, we have expanded on our discussion of the reasoning behind employing a non-reactive approach and the limitations that imposes on what questions can be answered in this study.

Without including the protonation as a CV, the authors tried to model the free energy changes from multiple FECs using different charge states of H87 and D342. This is a practical workaround, and the conclusion drawn (the OCC→ OF transition is downhill with protonated H87 and D342) seems valid. However, I don't think the OF states with different charge states (OF/H87D342-, OF/H87HD342-, OF/H87D342H, and OF/H87HD342H) are equally stable, as plotted in Figure 3b. The concern extends to other cases like Figures 4b, S7, S10, S12, S15, and S16. While it may be appropriate to match all four OF states in the free energy plot for comparison purposes, the authors should clarify this to ensure readers are not misled.

The reviewer is correct in their assessment that the aligning of PMFs in these figures is arbitrary; no relative free energies of the PMFs to each other can be estimated without explicit free energy calculations at least of protonation events at the end state basins. The PMFs in our figures are merely superimposed for illustrating the differences in shape between the obtained profiles in each condition, as discussed in the text, and we now make this clear in the appropriate figure captions.

b) Regarding the substrate impact, it appears that the authors assumed fixed protonation states. I am afraid this is not necessarily the case. Variations in PepT2 stoichiometry suggest that substrates likely participate in proton transport, like the Phe-Ala (2:1) and Phe-Gln (1:1) dipeptides mentioned in the introduction. And it is not rigorous to assume that the N- and C-termini of a peptide do not protonate/deprotonate when transported. I think the authors should explicitly state that the current work and the proposed mechanism (Figure 8) are based on the assumption that the substrates do not uptake/release proton(s).

This is indeed an assumption inherent in the current work. While we do “speculate that the proton movement processes may happen as an ensemble of different mechanisms, and potentially occur contemporaneously with the conformational change” we do not in the previous version indicate explicitly that this may involve the substrate. We make clear the assumption and this possibility in the revised version of our paper. Indeed, as we discuss, there is some evidence in our PMFs of an additional protonation site not considered thus far, which may or may not be the substrate. We now make note of this point in the revised manuscript.

As for what information can be drawn from the given experimental stoichiometries, we note in our paper that “a 2:1 stoichiometry was reported for the neutral di-peptide D-Phe-L-Ala and 3:1 for anionic D-Phe-L-Glu. (Chen et al., 1999) Alternatively, Fei et al. (1999) have found 1:1 stoichiometries for either of D-Phe-L-Gln (neutral), D-Phe-L-Glu (anionic), and D-Phe-L-Lys (cationic).”

We do not assume that it is our place to arbit among the apparent discrepancies in the experimental data here, although we believe that our assumed 2:1 stoichiometry is additionally “motivated also by our computational results that indicate distinct and additive roles played by two protons in the conformational cycle mechanism”.

(2) I have more serious concerns about the CpHMD employed in the study.a) The CpHMD in AMBER is not rigorous for membrane simulations. The underlying generalized Born model fails to consider the membrane environment when updating charge states. In other words, the CpHMD places a membrane protein in a water environment to judge if changes in charge states are energetically favorable. While this might not be a big issue for peripheral residues of membrane proteins, it is likely unphysical for internal residues like the ExxER motif. As I recall, the developers have never used the method to study membrane proteins themselves. The only CpHMD variant suitable for membrane proteins is the membrane-enabled hybrid-solvent CpHMD in CHARMM. While I do not expect the authors to redo their CpHMD simulations, I do hope the authors recognize the limitations of their method.

We discuss the limitations of the AMBER CpHMD implementation in the revised version. However, despite that, we believe we have in fact provided sufficient grounds for our conclusion that substrate binding affects ExxER motif protonation in the following way.

In addition to CpHMD simulations, we establish the same effect via ABFE calculations, where the substrate affinity is different at the E56 deprotonated vs protonated protein. This was figure S20 before, though in the revised version we have moved this piece of validation into a new panel of figure 6 in the main text, since it becomes more important with the CpHMD membrane problem in mind. Since the ABFE calculations are conducted with an all-atom representation of the lipids and the thermodynamic cycle closes well, it would appear that if the chosen CpHMD method has a systematic error of significant magnitude for this particular membrane protein system, there may be the benefit of error cancellation. While the calculated absolute pKa values may not be reliable, the difference made by substrate binding appears to be so, as judged by the orthogonal ABFE technique.

Although the reviewer does “not expect the authors to redo their CpHMD simulations”, we consider that it may be helpful to the reader to share in this response some results from trials using the continuous, all-atom constant pH implementation that has recently become available in GROMACS (Aho et al 2022, https://pubs.acs.org/doi/10.1021/acs.jctc.2c00516) and can be used rigorously with membrane proteins, given its all-atom lipid representation.

Unfortunately, when trying to titrate E56 in this CpHMD implementation, we found few protonationstate transitions taking place, and the system often got stuck in protonation state–local conformation coupled minima (which need to interconvert through rearrangements of the salt bridge network involving slow side-chain dihedral rotations in E53, E56 and R57). Author response image 1 shows this for the apo OF state, Author response image 2 shows how noisy attempts at pKa estimation from this data turn out to be, necessitating the use of a hybrid-solvent method.

**Author response image 1. sa4fig1:** All-atom CpHMD simulations of apo-OF PepT2. Red indicates protonated E56, blue is deprotonated.

**Author response image 2. sa4fig2:** Difficulty in calculating the E56 pKa value from the noisy all-atom CpHMD data shown in Author response image 1.

b) It appears that the authors did not make the substrate (Ala-Phe dipeptide) protonatable in holosimulations. This oversight prevents a complete representation of ligand-induced protonation events, particularly given that the substrate ion pairs with hsPepT2 through its N- & C-termini. I believe it would be valuable for the authors to acknowledge this potential limitation.

In this study, we implicitly assumed from the outset that the substrate does not get protonated, which – as by way of response to the comment above – we now acknowledge explicitly. This potential limitation for the available mechanisms for proton transfer also applies to our investigation of the ExxER protonation states. In particular, a semi-grand canonical ensemble that takes into account the possibility of substrate C-terminus protonation may also sample states in which the substrate is protonated and oriented away from R57, thus leaving the ExxER salt bridge network in an apo-like state. The consequence would be that while the direction of shift in E56 pKa value will be the same, our CpHMD may overestimate its magnitude. It would thus be interesting to make the C-terminus protonatable for obtaining better quantitative estimates of the E56 pKa shift (as is indeed true in general for any other protein protonatable residue, though the effects are usually assumed to be negligible). We do note, however, that convergence of the CpHMD simulations would be much harder if the slow degree of freedom of substrate reorientation (which in our experience takes 10s to 100s of nanoseconds in this binding pocket) needs to be implicitly equilibrated upon protonation state transitions. We discuss such considerations in the revised paper.

**Reviewer #2 (Public Review):**
This is an interesting manuscript that describes a series of molecular dynamics studies on the peptide transporter PepT2 (SLC15A2). They examine, in particular, the effect on the transport cycle of protonation of various charged amino acids within the protein. They then validate their conclusions by mutating two of the residues that they predict to be critical for transport in cell-based transport assays. The study suggests a series of protonation steps that are necessary for transport to occur in Petp2. Comparison with bacterial proteins from the same family shows that while the overall architecture of the proteins and likely mechanism are similar, the residues involved in the mechanism may differ.Strengths:This is an interesting and rigorous study that uses various state-of-the-art molecular dynamics techniques to dissect the transport cycle of PepT2 with nearly 1ms of sampling. It gives insight into the transport mechanism, investigating how the protonation of selected residues can alter the energetic barriers between various states of the transport cycle. The authors have, in general, been very careful in their interpretation of the data.Weaknesses:Interestingly, they suggest that there is an additional protonation event that may take place as the protein goes from occluded to inward-facing but they have not identified this residue.

We have indeed suggested that there may be an additional protonation site involved in the conformational cycle that we have not been able to capture, which – as we discuss in our paper – might be indicated by the shapes of the OCC ↔ IF PMFs given in Figure S15. One possibility is for this to be the substrate itself (see the response to reviewer #1 above) though within the scope of this study the precise pathway by which protons move down the transporter and the exact ordering of conformational change and proton transfer reactions remains a (partially) open question. We acknowledge this, denote it with question marks in the mechanistic overview we give in Figure 8 and also “speculate that the proton movement processes may happen as an ensemble of different mechanisms, and potentially occur contemporaneously with the conformational change”.

Some things are a little unclear. For instance, where does the state that they have defined as occluded sit on the diagram in Figure 1a? - is it truly the occluded state as shown on the diagram or does it tend to inward- or outward-facing?

Figure 1a is a simple schematic overview intended to show which structures of PepT2 homologues are available to use in simulations. This was not meant to be a quantitative classification of states. Nonetheless, we can note that the OCC state we derived has extra- and intracellular gate opening distances (as measured by the simple CVs defined in the methods and illustrated in Figure 2a) that indicate full gate closure at both sides. In particular, although it was derived from the IF state via biased sampling, the intracellular gate opening distance in the OCC state used for our conformational change enhanced sampling was comparable to that of the OF state (ie, full closure of the gate), see Figure S2b and the grey bars therein. Therefore, we would schematically classify the OCC state to lie at the center of the diagram in Figure 1a. Furthermore, it is largely stable over triplicates of 1 μslong unbiased MD, where in 2/3 replicates the gates remain stable, and the remaining replicate there is partial opening of the intracellular gate (as shown in Figure 2 b/c under the “apo standard” condition). We comment on this in the main text by saying that “The intracellular gate, by contrast, is more flexible than the extracellular gate even in the apo, standard protonation state”, and link it to the lower barrier for transition to IF than to OF. We did this by saying that “As for the OCC↔OF transitions, these results explain the behaviour we had previously observed in the unbiased MD of Figure 2c.” We acknowledge this was not sufficiently clear and have added details to the latter sentence to help clarify better the nature of the occluded state.

The pKa calculations and their interpretation are a bit unclear. Firstly, it is unclear whether they are using all the data in the calculations of the histograms, or just selected data and if so on what basis was this selection done. Secondly, they dismiss the pKa calculations of E53 in the outward-facing form as not being affected by peptide binding but say that E56 is when there seems to be a similar change in profile in the histograms.

In our manuscript, we have provided two distinct analyses of the raw CpHMD data. Firstly, we analysed the data by the replicates in which our simulations were conducted (Figure 6, shown as bar plots with mean from triplicates +/- standard deviation), where we found that only the effect on E56 protonation was distinct as lying beyond the combined error bars. This analysis uses the full amount of sampling conducted for each replicate. However, since we found that the range of pKa values estimated from 10ns/window chunks was larger than the error bars obtained from the replicate analysis (Figures S17 and S18), we sought to verify our conclusion by pooling all chunk estimates and plotting histograms (Figure S19). We recover from those the effect of substrate binding on the E56 protonation state on both the OF and OCC states. However, as the reviewer has pointed out (something we did not discuss in our original manuscript), there is a shift in the pKa of E53 of the OF state only. In fact, the trend is also apparent in the replicate-based analysis of Figure 6, though here the larger error bars overlap. In our revision, we added more details of these analyses for clarity (including more detailed figure captions regarding the data used in Figure 6) as well as a discussion of the partial effect on the E53 pKa value.

We do not believe, however, that our key conclusions are negatively affected. If anything, a further effect on the E53 pKa which we had not previously commented on (since we saw the evidence as weaker, pertaining to only one conformational state) would strengthen the case for an involvement of the ExxER motif in ligand coupling.

**Reviewer #3 (Public Review):**
Summary:Lichtinger et al. have used an extensive set of molecular dynamics (MD) simulations to study the conformational dynamics and transport cycle of an important member of the proton-coupled oligopeptide transporters (POTs), namely SLC15A2 or PepT2. This protein is one of the most wellstudied mammalian POT transporters that provides a good model with enough insight and structural information to be studied computationally using advanced enhanced sampling methods employed in this work. The authors have used microsecond-level MD simulations, constant-PH MD, and alchemical binding free energy calculations along with cell-based transport assay measurements; however, the most important part of this work is the use of enhanced sampling techniques to study the conformational dynamics of PepT2 under different conditions.The study attempts to identify links between conformational dynamics and chemical events such as proton binding, ligand-protein interactions, and intramolecular interactions. The ultimate goal is of course to understand the proton-coupled peptide and drug transport by PepT2 and homologous transporters in the solute carrier family.Some of the key results include:(1) Protonation of H87 and D342 initiate the occluded (Occ) to the outward-facing (OF) state transition.(2) In the OF state, through engaging R57, substrate entry increases the pKa value of E56 and thermodynamically facilitates the movement of protons further down.(3) E622 is not only essential for peptide recognition but also its protonation facilitates substrate release and contributes to the intracellular gate opening. In addition, cell-based transport assays show that mutation of residues such as H87 and D342 significantly decreases transport activity as expected from simulations.Strengths:(1) This is an extensive MD-based study of PepT2, which is beyond the typical MD studies both in terms of the sheer volume of simulations as well as the advanced methodology used. The authors have not limited themselves to one approach and have appropriately combined equilibrium MD with alchemical free energy calculations, constant-pH MD, and geometry-based free energy calculations. Each of these 4 methods provides a unique insight regarding the transport mechanism of PepT2.(2) The authors have not limited themselves to computational work and have performed experiments as well. The cell-based transport assays clearly establish the importance of the residues that have been identified as significant contributors to the transport mechanism using simulations.(3) The conclusions made based on the simulations are mostly convincing and provide useful information regarding the proton pathway and the role of important residues in proton binding, protein-ligand interaction, and conformational changes.Weaknesses:(1) Some of the statements made in the manuscript are not convincing and do not abide by the standards that are mostly followed in the manuscript. For instance, on page 4, it is stated that "the K64-D317 interaction is formed in only ≈ 70% of MD frames and therefore is unlikely to contribute much to extracellular gate stability." I do not agree that 70% is negligible. Particularly, Figure S3 does not include the time series so it is not clear whether the 30% of the time where the salt bridge is broken is in the beginning or the end of simulations. For instance, it is likely that the salt bridge is not initially present and then it forms very strongly. Of course, this is just one possible scenario but the point is that Figure S3 does not rule out the possibility of a significant role for the K64-D317 salt bridge.

The reviewer is right to point out that the statement and Figure S3 as they were do not adequately support our decision to exclude the K64-D317 salt-bridge in our further investigations. The violin plot shown in Figure S3, visualised as pooled data from unbiased 1 μs triplicates, did indeed not rule out a scenario where the salt bridge only formed late in our simulations (or only in some replicates), but then is stable. Therefore, in our revision, we include the appropriate time-series of the salt bridge distances, showing how K64-D317 is initially stable but then falls apart in replicate 1, and is transiently formed and disengaged across the trajectories in replicates 2 and 3. We have also remade the data for this plot as we discovered a bug in the relevant analysis script that meant the D170-K642 distance was not calculated accurately. The results are however almost identical, and our conclusions remain.

(2) Similarly, on page 4, it is stated that "whether by protonation or mutation - the extracellular gate only opens spontaneously when both the H87 interaction network and D342-R206 are perturbed (Figure S5)." I do not agree with this assessment. The authors need to be aware of the limitations of this approach. Consider "WT H87-prot" and "D342A H87-prot": when D342 residue is mutated, in one out of 3 simulations, we see the opening of the gate within 1 us. When D342 residue is not mutated we do not see the opening in any of the 3 simulations within 1 us. It is quite likely that if rather than 3 we have 10 simulations or rather than 1 us we have 10 us simulations, the 0/3 to 1/3 changes significantly. I do not find this argument and conclusion compelling at all.

If the conclusions were based on that alone, then we would agree. However, this section of work covers merely the observations of the initial unbiased simulations which we go on to test/explore with enhanced sampling in the rest of the paper, and which then lead us to the eventual conclusions.

Figure S5 shows the results from triplicate 1 μs-long trajectories as violin-plot histograms of the extracellular gate opening distance, also indicating the first and final frames of the trajectories as connected by an arrow for orientation – a format we chose for intuitively comparing 48 trajectories in one plot. The reviewer reads the plot correctly when they analyse the “WT H87-prot” vs “D342A H87-prot” conditions. In the former case, no spontaneous opening in unbiased MD is taking place, whereas when D342 is mutated to alanine in addition to H87 protonation, we see spontaneous transition in 1 out of 3 replicates. However, the reviewer does not seem to interpret the statement in question in our paper (“the extracellular gate only opens spontaneously when both the H87 interaction network and D342-R206 are perturbed”) in the way we intended it to be understood. We merely want to note here a correlation in the unbiased dataset we collected at this stage, and indeed the one spontaneous opening in the case comparison picked out by the reviewer is in the condition where both the H87 interaction network and D342-R206 are perturbed. In noting this we do not intend to make statistically significant statements from the limited dataset. Instead, we write that “these simulations show a large amount of stochasticity and drawing clean conclusions from the data is difficult”. We do however stand by our assessment that from this limited data we can “already appreciate a possible mechanism where protons move down the transporter pore” – a hypothesis we investigate more rigorously with enhanced sampling in the rest of the paper. We have revised the section in question to make clearer that the unbiased MD is only meant to give an initial hypothesis here to be investigated in more detail in the following sections. In doing so, we also incorporate, as we had not done before, the case (not picked out by the reviewer here but concerning the same figure) of S321A & H87 prot. In the third replicate, this shows partial gate opening towards the end of the unbiased trajectory (despite D342 not being affected), highlighting further the stochastic nature that makes even clear correlative conclusions difficult to draw.

(3) While the MEMENTO methodology is novel and interesting, the method is presented as flawless in the manuscript, which is not true at all. It is stated on Page 5 with regards to the path generated by MEMENTO that "These paths are then by definition non-hysteretic." I think this is too big of a claim to say the paths generated by MEMENTO are non-hysteretic by definition. This claim is not even mentioned in the original MEMENTO paper. What is mentioned is that linear interpolation generates a hysteresis-free path by definition. There are two important problems here: (a) MEMENTO uses the linear interpolation as an initial step but modifies the intermediates significantly later so they are no longer linearly interpolated structures and thus the path is no longer hysteresisfree; (b) a more serious problem is the attribution of by-definition hysteresis-free features to the linearly interpolated states. This is based on conflating the hysteresis-free and unique concepts. The hysteresis in MD-based enhanced sampling is related to the presence of barriers in orthogonal space. For instance, one may use a non-linear interpolation of any type and get a unique pathway, which could be substantially different from the one coming from the linear interpolation. None of these paths will be hysteresis-free necessarily once subjected to MD-based enhanced sampling techniques.

We certainly do not intend to claim that the MEMENTO method is flawless. The concern the reviewer raises around the statement "These paths are then by definition non-hysteretic" is perhaps best addressed by a clarification of the language used and considering how MEMENTO is applied in this work.

Hysteresis in the most general sense denotes the dependence of a system on its history, or – more specifically – the lagging behind of the system state with regards to some physical driver (for example the external field in magnetism, whence the term originates). In the context of biased MD and enhanced sampling, hysteresis commonly denotes the phenomenon where a path created by a biased dynamics method along a certain collective variable lags behind in phase space in slow orthogonal degrees of freedom (see Figure 1 in Lichtinger and Biggin 2023, https://doi.org/10.1021/acs.jctc.3c00140). When used to generate free energy profiles, this can manifest as starting state bias, where the conformational state that was used to seed the biased dynamics appears lower in free energy than alternative states. Figure S6 shows this effect on the PepT2 system for both steered MD (heavy atom RMSD CV) + umbrella sampling (tip CV) and metadynamics (tip CV). There is, in essence, a coupled problem: without an appropriate CV (which we did not have to start with here), path generation that is required for enhanced sampling displays hysteresis, but the refinement of CVs is only feasible when paths connecting the true phase space basins of the two conformations are available. MEMENTO helps solve this issue by reconstructing protein conformations along morphing paths which perform much better than steered MD paths with respect to giving consistent free energy profiles (see Figure S7 and the validation cases in the MEMENTO paper), even if the same CV is used in umbrella sampling.

There are still differences between replicates in those PMFs, indicating slow conformational flexibility propagated from end-state sampling through MEMENTO. We use this to refine the CVs further with dimensionality reduction (see the Method section and Figure S8), before moving to 2D-umbrella sampling (figure 3). Here, we think, the reviewer’s point seems to bear. The MEMENTO paths are ‘non-hysteretic by definition’ with respect to given end states in the sense that they connect (by definition) the correct conformations at both end-states (unlike steered MD), which in enhanced sampling manifests as the absence of the strong starting-state bias we had previously observed (Figure S7 vs S6). They are not, however, hysteresis-free with regards to how representative of the end-state conformational flexibility the structures given to MEMENTO really were, which is where the iterative CV design and combination of several MEMENTO paths in 2D-PMFs comes in.

We also cannot make a direct claim about whether in the transition region the MEMENTO paths might be separated from the true (lower free energy) transition paths by slow orthogonal degrees of freedom, which may conceivably result in overestimated barrier heights separating two free energy basins. We cannot guarantee that this is not the case, but neither in our MEMENTO validation examples nor in this work have we encountered any indications of a problem here.

We hope that the reviewer will be satisfied by our revision, where we replace the wording in question by a statement that the MEMENTO paths do not suffer from hysteresis that is otherwise incurred as a consequence of not reaching the correct target state in the biased run (in some orthogonal degrees of freedom).

**Recommendations for the authors:**

**Reviewer #2 (Recommendations For The Authors):**
Figure S1: it would be useful to label the panels.

We have now done this.

At the bottom of page 4, it is written that "the extracellular gate only opens spontaneously when both the H87 interaction network and D342-R206 are perturbed (Figure S5)." But it is hard to interpret that from the figure.

See also our response to reviewer #3. We have revised the wording of this statement, and also highlight in Figure S5 the crucial runs we are referring to, in order to make them easier to discern.

At the bottom of page 5, and top of page 6, there is a lot of "other" information shown, which is inserted for the record - this is a bit glossed over and hard to follow.

The “other” information refers to further conditions we had calculated PMFs for and that gave some insight, but which were secondary for drawing our key conclusions. We thank the reviewer for their feedback that this section needs clarification. We have revised this paragraph to make it easier to follow and highlight better the conclusions we draw form the data.

In Figure 7 it looks as though the asterisks have shifted.

We are indebted to the reviewer for spotting this error, the asterisks are indeed shifted one bar to the right of their intended position. The revised version fixes this issue.

**Reviewer #3 (Recommendations For The Authors):**
Minor points: In Figure 1a, The 7PMY label and arrow are slightly misplaced.

Figure 1a is a schematic diagram to show the available structures of PepT2 homologues (see also the response to reviewer #2 above). The 7PMY label placement is intentional to indicate a partially occluded inwards-facing state. As we write in the figure caption: “Intermediate positions between states indicate partial gate opening”.